# Verified Code Transpilation with LLMs

**Sahil Bhatia**[1]    **Jie Qiu**    **Niranjan Hasabnis**[2]*    **Sanjit A. Seshia**[1]    **Alvin Cheung**[1]
[1]UC Berkeley    [2]Code Metal
{sahilbhatia, jieq, sseshia, akcheung}@berkeley.edu
niranjan@codemetal.ai

## Abstract

Domain-specific languages (DSLs) are integral to various software workflows. Such languages offer domain-specific optimizations and abstractions that improve code readability and maintainability. However, leveraging these languages requires developers to rewrite existing code using the specific DSL's API. While large language models (LLMs) have shown some success in automatic code transpilation, none of them provide any functional correctness guarantees on the transpiled code. Another approach for automating this task is verified lifting, which relies on program synthesis to find programs in the target language that are functionally equivalent to the source language program. While several verified lifting tools have been developed for various application domains, they are specialized for specific source-target languages or require significant expertise in domain knowledge to make the search efficient. In this paper, leveraging recent advances in LLMs, we propose an LLM-based approach (LLMLIFT) to building verified lifting tools. We use the LLM's capabilities to reason about programs to translate a given program into its corresponding equivalent in the target language. Additionally, we use LLMs to generate proofs for functional equivalence. We develop lifting-based compilers for *four different* DSLs targeting different application domains. Our approach not only outperforms previous symbolic-based tools in both the number of benchmarks transpiled and transpilation time, but also requires significantly less effort to build.

## 1 Introduction

Domain-specific languages (DSLs) have gained popularity due to their ability to provide optimizations and abstractions that improve code readability and performance in specific domains. Examples of recent DSLs include Spark (distributed computing), NumPy (array processing), TACO (tensor processing), and P4 (network packet processing). With new DSLs emerging for diverse application domains and programming languages, developers often face the task of manually rewriting existing code to incorporate these languages into their existing workflows. This manual rewriting process can be tedious, may introduce bugs into the code, and may fail to preserve the semantics of the starting code. This problem of transforming and compiling code from one programming language to another is called *transpilation*. The question we address in this paper is: can large language models (LLMs) correctly and automatically perform code transpilation?

A particularly useful form of code transpilation, termed *lifting*, involves translating code in a somewhat lower-level, general-purpose language to equivalent code in a DSL. Lifting allows developers to port code to DSLs from which efficient code can be generated for special-purpose hardware, such as GPUs, machine learning accelerators, or network processors. Therefore, significant effort has been dedicated to developing tools aimed at automating the task of lifting. Rule-based approaches rely on traditional pattern-matching techniques [1]; however, describing these rules can be a complex,

---

*Work was done while at Intel Labs.

human-intensive task. An alternative are search-based techniques that leverage advances in *program synthesis* (e.g., see [2, 3, 4]) and formal verification over the last two decades. The use of verified program synthesis for lifting, termed *verified lifting*, involves searching for a program in the DSL and subsequently formally verifying its semantic equivalence to the source program. Verified lifting has been successfully applied in building compilers [5, 6, 7, 8] for DSLs like Spark, SQL, Halide, and TACO. Contemporary program synthesis approaches can be broadly classified into two categories: *symbolic* and *neural*. Traditionally, symbolic techniques such as enumerative, deductive, and constraint-based synthesis strategies have been used for implementing the search. More recently, neural networks [9] have been trained and leveraged to accelerate the search process. Despite their successes, both symbolic and neural approaches have common drawbacks: 1) The synthesizer is customized for each DSL, making them challenging to adapt for new DSLs, and 2) Significant effort is required to design the synthesizer, including domain-specific heuristics for symbolic approaches and the generation of parallel corpora $\langle source, target \rangle$ for ML-based approaches, to enable generalization and scalability for the target DSL.

Large Language Models (LLMs) [10, 11] have emerged as a promising approach for tackling complex programming tasks, including code generation, repair, and testing. However, generating reliable code with formal correctness guarantees with LLMs remains challenging. Most work on LLMs either focuses on generating code without correctness guarantees [12, 13, 14] or separately on producing proof annotations (such as invariants) for given code [15, 16]. Additionally, formal verification tools often have their own specialized languages (e.g., SMT-LIB, Dafny) for encoding verification problems and specifications. These languages are typically low-resource in the training datasets of LLMs, making it challenging for the models to generate code in these formal verification languages directly. To leverage LLMs for building verified lifting compilers, we must address two key constraints: generalization to new DSLs and providing correctness guarantees for the generated code.

In this work, we investigate the use of *LLMs for verified lifting (VL)*. Our approach, called LLMLIFT, takes inspiration from the core technique of VL, which involves translating the source program to a higher-level intermediate representation (IR) that describes the semantics of the DSL operators. Once the synthesized code is verified, it is then translated to the concrete syntax of the DSL using rewrite rules. We leverage the reasoning capabilities of LLMs to translate code from context to an IR. We instruct the model via a prompt to generate code using the operators of the DSL, with Python serving as the IR to encode the semantics of these operators. Python's significant representation in the training datasets of LLMs makes it a suitable choice for this purpose. In addition to generating the DSL program, we also prompt the model to generate a proof of correctness for the program. To the best of our knowledge, our approach is the first to leverage LLMs to generate *both code and proof annotations* together. To verify the functional equivalence of the generated program to the given source program for all program states, we translate both the generated program and the proof to the syntax of an automated theorem prover. This step ensures that the synthesized code is formally verified and can be trusted to be correct. Our evaluation (section Sec. 3) shows that LLMLIFT has significant advantages over traditional search-based symbolic VL-based tools. It solves **7** more benchmarks, requires substantially less effort in terms of LoC (**1000**×), and is faster in generating verified code and proofs (**6**× on average).

In summary, this paper makes the following novel contributions

1. We introduce the first technique for formally-verified code transpilation using LLMs.
2. Our approach uses Python as an IR for code generation, thus eliminating the need for specialized DSL-specific training data or fine-tuning of LLMs.
3. Our method eliminates the need for manual encoding of domain-specific heuristics, thus simplifying the process of verified lifting by reducing the human effort required in traditional techniques.
4. We propose an approach to generate not only the lifted code but also a proof of correctness for the generated code. This integration of LLMs with verification oracles guarantees the correctness of the generated code, a crucial aspect that sets our approach apart from other work on LLM-based code generation.
5. We show the effectiveness of our approach (Sec. 4) by constructing compilers for **four** DSLs spanning various application domains. In terms of accuracy, our LLM-based compilers achieve comparable performance to existing tools and, in some domains, outperforms the prior approaches.

## 2 Background

```
1  def matrix_add(a: List[List[int]], b: List[List[int]])
2  -> List[List[int]]:
3    return (
4      []
5      if len(a) < 1
6      or not len(a) == len(b)
7      or vec_add(a[0], b[0]) ==[]
8      else [
9        vec_add(a[0], b[0]),
10       *matrix_add(a[1:],b[1:]),
11     ])
12 def matrix_scalar_sub(a: int, b: List[List[int]])
13 -> List[List[int]]:
14   return (
15     []
16     if len(b) < 1
17     else [
18       vec_scalar_sub(a, b[0]),
19       *matrix_scalar_sub(a, b[1:])
20     ])
```

```
1  vector<vector<int>> test(
2    vector<vector<int>> b,
3    vector<vector<int>> a) {
4    vector<vector<int>> out;
5    for (int row = 0; row < b.size(); row++) {
6      vector<int> row_vec;
7      for (int col = 0; col < b[0].size(); col++) {
8        int pixel = b[row][col] + a[row][col] - 255;
9        row_vec.push_back(pixel);}
10     out.push_back(row_vec);}
11   return out;}
```

(a) Source Code (S).    (b) Target Language ($T_{lang}$).

Figure 1: Sequential source code in C++ and semantics of DSL in IR.

We now give an overview and an end-to-end example of verified lifting (VL) where we use program synthesis to build a compiler. Given a program (S) in the source language ($S_{lang}$), VL uses a search procedure to find a program (T) in the target language ($T_{lang}$) that can be proved to be functionally equivalent to the given source program. VL comprises of three phases: 1) Search, 2) Verification, and 3) Code generation. The key behind VL is to first transpile S to an user-defined intermediate representation (IR) of the operators in the target language before generating executable code. The IR serves as a *functional description* of $T_{lang}$ and ignores any implementation details. Hence, during search phase, S is **lifted** to a sequence of operators expressed using the IR. This expression serves as the program summary ($PS$) which summarizes S using the IR. Subsequently, $PS$ is **verified** using a theorem prover to check for semantic equivalence with S for all program inputs. If verification succeeds, $PS$ is then translated into the concrete syntax of the target language using simple pattern-matching rules provided by the user to generate executable code. These rules are notably simpler to write compared to a rule-based translator that directly compiles from $S_{lang}$ to $T_{lang}$, as the $PS$ is already expressed using the operators in the target language.

We demonstrate an example of transpiling a sequential C++ to tensor-processing frameworks (such as PyTorch, Tensorflow, NumPy) using VL. Tensor-processing frameworks provide high-level API for performing, large-scale numerical computation on multi-dimensional arrays. Some of the basic tensor operators supported by all the frameworks are elementwise operators (tensor-tensor, tensor-scalar).

Fig. 1a shows a sequential source program (S) performing the linear burn blending operation in image editing. The given source program takes as input two images (represented as 2D vectors) and processes each pixel from both the images by first adding them and then subtracting by integer 255.

In Fig. 1b, we define the semantics of the tensor operators such as `matrix_add` and `matrix_scalar_sub`. These functions abstract the implementation details of the operators in the tensor-processing frameworks while only capturing the high-level semantics of the operators. Our goal is to find an IR expression sequence of these operators such that it is semantically equivalent to S. Traditional approaches to solving this search problem in VL involve framing it as SyGuS [17] problem. SyGuS is an approach for solving program synthesis problems by specifying constraints and searching for solutions within a defined space. Specifically, a SyGuS problem involves defining a search space that syntactically restricts the space of possible solutions, thereby making the search tractable. Formally, this objective can be stated as $\exists\, T \in T_{lang} \mid \forall\, \sigma.\, S(\sigma) = T(\sigma)$, where T is a program in the target language. For our program in Fig. 1a, the synthesis phase would return the following $PS$ (i.e., T):

```
matrix_scalar_sub(255, matrix_add(b, a))
```

This expression performs element-wise addition of two matrices a and b, followed by a scalar subtraction of 255 from each element of the resulting matrix. Since S contains a loop, proving equivalence with the generated program requires another predicate called the "loop invariant." A loop invariant is a logical statement that must hold before and after each iteration of a loop. Intuitively, it

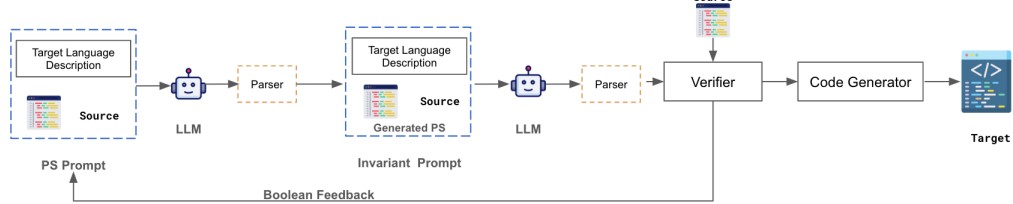

Figure 2: A high-level overview of our LLMLIFT framework for building verified lifting-based tools.

captures the essential properties that are preserved while the loop executes. During VL's synthesis phase, we generate both the program summary and any required loop invariant for verification. Verification is done by sending the program summary and loop invariant(s) to a theorem prover. Verifier checks the semantic equivalence between S and the generated program summaries. VL currently uses cvc5 and z3 for this purpose. Once verified, we translate the generated program summary to the concrete syntax of the DSL (NumPy) using simple pattern-matching rules, resulting in the following executable code:

```
def linear_burn_8_np(b: np.ndarray, a: np.ndarray) -> np.ndarray:
  return a + b - 255
```

We next describe how our LLM-based approach can improve VL synthesis problem.

## 3 LLM-Based Verified Lifting

We now describe our LLM-based approach for verified lifting. We begin by formalizing the VL problem. Then we give details of how we use LLMs to improve over the traditional VL approach.

### 3.1 Problem Formulation

The VL problem is characterized by three inputs:

1. **Specification** ($\phi$): The specification ($\phi$) defines the property that the target program (T) should satisfy. For VL considered in this paper, the source and target programs are *side-effect free functions* of their inputs. Thus, $\phi$ encodes the semantic equivalence of T and the source program (S) for each program input state $\sigma$. The overall correctness condition is:

$$\forall \sigma \; \phi(\sigma, \texttt{T}, \texttt{S}) \; \doteq \; \forall \, \sigma. \, \texttt{S}(\sigma) = \texttt{T}(\sigma) \tag{1}$$

2. **Program Space** ($G$): The program space outlines the set of potential solutions, typically expressed as a context-free grammar $G$. The language of $G$ includes all sequences of operators $ops \in T_{lang}$ applied recursively to terms starting with input variables $\sigma$. A target program T is a program summary $PS$ that is a composition of operators $ops$. An example involving the tensor operators is provided in the previous section. In other words, for each input $\sigma$, S ($\sigma$) must be expressed using a combination of operators ($ops$) from $T_{lang}$.

3. **Certificate Space:** ($G_I$) A key part of the VL problem is to generate a certificate of correctness, typically in the form of invariants that a verifier can use to prove that $\phi$ holds for the generated $PS$. Synthesis tools typically use a grammar $G_I$ to constrain the space of possible invariants to search over; we refer to this as the *certificate space*.

**VL Problem:** Given the inputs S, $\phi$, $G$, $G_I$, and $T_{lang}$ described above, the VL problem is to generate a correct target program T in $T_{lang}$ represented as the combination ($PS$, $Inv$). This is formally expressed in logic as follows:

$$\exists \, PS \in G \;\; \exists Inv \in G_I \;\; \forall \sigma \, . \, \phi(\sigma, (PS, Inv), \texttt{S}) \tag{2}$$

This states that we aim to find a program summary ($PS$) and invariants ($Inv$) from the defined search space $G, G_I$ such that the given specification (functional equivalence with S) holds for all possible program states $\sigma$.

Traditionally, the VL problem has been solved by symbolic program synthesis solvers utilizing methods such as enumerative search, deductive search, and constraint-based approaches [17, 4]. These

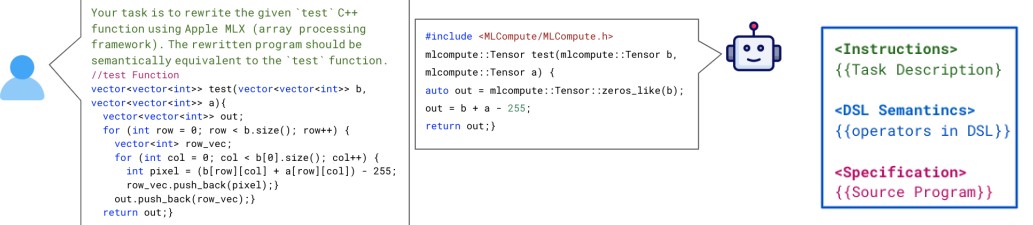

Figure 3: End-to-End Lifting Example.

Figure 4: Prompt Structure.

rely heavily on manually-designed heuristics to make the search over the program and certificate space effective. Unfortunately doing so is resource-intensive and requires domain-specific knowledge. To address these limitations, we take a new approach that leverages LLMs.

## 3.2 LLM-based Verified Lifting

A naive approach to building a VL-based compiler using LLMs would be to prompt LLMs to translate $S_{lang}$ programs directly into $T_{lang}$. However, this approach has the following shortcomings:

1. VL-based compilers require that the $T_{lang}$ candidates generated during the search phase be functionally equivalent to the input $S_{lang}$ program. This is a strong requirement that current LLMs are unable to satisfy on their own.

2. Unlike widely-used general-purpose languages such as Python, domain-specific languages (DSLs) are used only in their niche applications. Unsurprisingly, we find that LLMs struggle to generate code in languages that are insufficiently represented in their training data.

In Fig. 3, we show an example of LLMs struggling to generate code reliably in new DSLs. We instruct GPT-4o to translate the program in Fig. 1a to MLX (Apple's latest tensor processing DSL), and the model fails to generate the expected python MLX code. Instead, the model outputs a completely incorrect solution by hallucinating non-existent `MLCompute` header file. This problem is even more prominent for new DSLs that the model might have never seen in the training dataset.

**Our Approach:** To address these challenges, we propose an approach that adapt two key ideas from traditional synthesis to the LLM setting:

1. *Python as an Intermediate Representation (IR):* We adopt VL's key idea of *transpiling to an IR rather than directly to the concrete syntax of $T_{lang}$.* Specifically, since Python is highly represented in the training dataset of popular LLMs [12], these LLMs are effective at generating syntactically-correct Python code. We exploit these observations by leveraging Python as the IR to define semantics of DSL operators. An example is shown in Fig. 1b.

2. *Oracle-Guided Inductive Synthesis (OGIS) with LLM oracles:* Traditional verified lifting follows the paradigm of counterexample-guided inductive synthesis (CEGIS) [2], where a Learner that synthesizes from examples interacts with a verifier that checks the Learner's output programs for correctness with respect to a specification. OGIS [18, 19] is a generalization of CEGIS with a richer *oracle interface* that allows for more expressive oracles and interactions. Our LLM Lifting approach instantiates OGIS with LLM oracles that synthesize $PS$ and $Inv$ before invoking a verifier to check correctness. Our queries to the LLM oracles (prompts) follow a few-shot learning approach. Our current verifier provides Boolean feedback to the LLM oracles.

In Fig. 2, we show our LLM-based OGIS approach, where LLMs are applied to generate program summaries and invariants in the IR, using a few-shot learning framework that we describe next.

## 3.3 Few-Shot Learning Approach

LLMs have demonstrated few-shot reasoning capabilities [11]. Few-shot reasoning allows LLMs to generalize their understanding to new tasks by leveraging a small set of similar examples. Enabling them to extend their reasoning capabilities to tasks without requiring explicit training or fine-tuning

for those specific tasks. We propose leveraging the few-shot reasoning capabilities of LLMs for verified lifting as fine-tuning existing LLMs for each new DSL is often infeasible due to the lack of extensive training data and the rapid pace at which new DSLs are developed. The effort required to collect, annotate, and preprocess DSL-specific training data for fine-tuning can be substantial, making it impractical to adapt LLMs to each new DSL.

As described in Sec. 2, in VL, we generate candidates in an IR that abstracts away low-level implementation details of the operators in $T_{lang}$. The objective, as defined in Eq. (2), is to find $PS$ and $Inv$ expressed using operators from $T_{lang}$ such that $\phi$ holds. We leverage the few-shot reasoning capability by providing the models with the semantics of operators from the target language ($T_{lang}$) using an IR. By exposing the LLMs to these semantics, we enable them to use their reasoning capabilities over code to generate both the $PS$ and invariants in the IR.

In Fig. 4, we illustrate the high-level prompt structure we use to generate the $PS$ and $Inv$. The prompt consists of the following components:

1. **Task Instruction.** We instruct the model using a natural language to translate S using only the specified DSL operators.
2. **DSL Operators.** We specify the semantics of all operators from $T_{lang}$ using Python and include it in the prompt. Python is chosen as our IR due to (a) its widespread use across domains, (b) its concise and expressive nature, making the representation readable and straightforward and, (c) its significant representation in code datasets used for training LLMs [12].
3. **Specification.** While symbolic techniques often rely on approaches like test cases, bounded model checking, and Hoare logic [20] for defining specifications, the natural language interface of LLMs offers flexibility in using various specifications and combining different forms. Given that LLMs are primarily trained on raw source code and may not have encountered other forms of specification during training, we directly use the source program (S) as the specification in our prompt.

We next describe the end-to-end workflow for our LLM-based verified lifting.

**PS Guessing.** We split the generation of $PS$ and $Inv$ into a two-phase process by first asking the LLM to generate the $PS$ and then inferring invariants corresponding to it. For generating $PS$, we prompt the model in zero-shot setting. Due to space constraints, we show an instantiation of the prompt structure shown in Fig. 4 in Appendix B. When prompted, the model generates the following $PS$ for our example code shown in Fig. 1a, representing the S as a combination of DSL operators:

```
matrix_scalar_sub(255, matrix_add(b, a))
```

To ensure that the generated candidates follow the DSL operators defined in the prompt, we use a rule-based parser to reject any candidates that do not satisfy this constraint, i.e., those that use constructs outside the DSL operators (see Appendix D for examples).

**Inv Guessing**. Next, if S contains loops, establishing the functional equivalence of the generated $PS$ for all program states with S requires loop invariants. In VL, loop invariants typically follow a templated structure:

$$Inv \triangleq f(i) \land e(T_{lang}) \tag{3}$$

where $f(i)$ denotes an expression over loop indexes and $e(T_{lang})$ represents an inductive expression constructed using operators from $T_{lang}$. This structured nature simplifies the invariant generation process compared to solving general loop invariant synthesis problems. To facilitate the generation of loop invariants, we use one-shot learning (unlike the zero-shot approach for program summaries). This is needed: 1. to familiarize the model with the concept and structure of invariants in the VL context and, 2. generating program summaries is relatively easier than loop invariants, as the model's primary instruction is simply to combine operators from the given DSL without introducing external ones—a constraint that is easily expressed in natural language (due to space constraints we illustrate the prompt in Appendix B)[2]. The prompt for invariant generation closely resembles that used for generating program summaries, including S with an additional assertion stating the equality of the return variable with the previously generated $PS$. This instruction guides the model to produce an invariant corresponding to the generated $PS$. The invariants are generated as Boolean expressions in Python rather than SMT-LIB, as we found that LLMs encounter difficulties in generating SMT-LIB (standard format for SMT solvers) code due to its limited representation in training datasets. When prompted, model generates the following invariant for the code shown in Fig. 1a:

---

[2]We evaluate LLMLIFT with a zero-shot invariant prompt in Appendix K.

```
def invariant_outer(row, col, b, a, out):
  return row >= 0 and row <= len(b) and
    out == matrix_scalar_sub(255, matrix_add(b[:i], a[:i]))
```

The loop invariant states that the loop index $row$ remains within the bounds of the array $b$ (i.e., $0 \leq i \leq \text{len}(b)$). Additionally, the invariant expresses $out$ as a tensor DSL expression over the first $i$ elements of the inputs $b$ and $a$, which helps verify that the invariant holds in each iteration of the loop. Similar to $PS$ generation, the generated loop invariants are also checked using our rule-based parser to ensure they conform to the DSL.

**Verification**. Both the generated $PS$ and $Inv$ are expressed in Python. We use simple pattern-matching rewrite rules to translate these expressions into syntax compatible with the verification oracle, which checks for functional equivalence. The objective is to verify that the given S and generated T are equivalent for all possible inputs to the S program. In Appendix E, we provide a proof demonstrating how we establish this functional equivalence using an SMT solver. If the verifier cannot establish validity, we start the process again to generate new candidates.

**Code Generation.** Once verified, the $PS$ is translated into the concrete syntax of $T_{lang}$ using straightforward rewrite rules that recursively parse the generated $PS$ and translate it into the concrete operators of the DSL, leveraging the syntactic nature of Python. The translation process is simplified due to Python's highly structured syntax. For instance, the generated $PS$ for our running example can be translated into tensor processing frameworks like NumPy, generating the following code:

```
def linear_burn_8_np(b, a):
    return a + b - 255
```

See Appendix F for more details on rule-based code generator. We present our complete algorithm for generating $PS$ and $Inv$ in Appendix A.

## 4 Experiments

To evaluate the effectiveness of LLMLIFT, we evaluate across four distinct DSLs[3], each targeting a different application domain:

1. **Distributed Computing**: We transpile sequential Java programs into MapReduce implementations written using the Apache Spark [21] API. Spark, an open-source distributed computing framework, provides an interface for programming multiple clusters for data parallelism which helps in large-scale data processing.
2. **Network Packet Processing**: We transpile sequential network processing algorithms in C to the operators of programmable switch devices [22] with its own ISA. This translation enables the exploration of novel algorithms, such as congestion control and load balancing, on programmable switch devices.
3. **TACO**: We transpile sequential C++ programs into TACO [23]'s API. TACO is a tensor processing compiler for generating highly optimized GPU code for performing tensor computations.
4. **Tensor Processing**. We transpile sequential C++ programs to a recently introduced tensor processing IR called TensIR [24]. TensIR consists of common tensor operations such as element-wise arithmetic operators, reduction operators and transpose, among others. TensIR is designed to enable translation of unoptimized sequential code to tensor operations which can be then executed on 6 different software and hardware backends.

**Implementation Details**: In all experiments, we use GPT-4 via their APIs to generate candidates. We set the temperature to 0.7 for all the experiments. For program summary and invariant generation across all domains, we use the same zero-shot $PS$ prompt in Fig. 6 and one-shot prompt in Fig. 7, respectively. We keep a budget of 50 queries for the $PS$ and a budget of 10 queries for each $PS$. The parser and logic for invoking the LLMs are implemented in Python with $\approx$700 LoC.

**Note**. LLMLift currently only supports a subset of the C/C++ and Python language in the source programs. In particular, it does not support any code that uses pointers or objects as verifying programs with these constructs is challenging. That said, we did not encounter the use of these constructs in any of the benchmarks in all the four domains that we evaluated on.

---

[3]All the benchmarks used for evaluation can be found at:
https://github.com/metalift/metalift/tree/llmlift/llmlift/benchmarks

We present the results in the sections below and defer the error analysis to Appendix D. We also provide an analysis on the performance of the generated code in Appendix G.

## 4.1 Distributed Computing

MapReduce, a programming model for parallel processing of large datasets across distributed clusters, simplifies parallel computation by abstracting away distributed system complexities. A MapReduce program comprises two phases: 1. Map: Input data is partitioned into smaller chunks, each processed by a mapper function to generate key-value pairs. 2. Reduce: Intermediate key-value pairs are shuffled, sorted based on keys, and then processed by reducer functions to aggregate associated values.

**LLMLIFT implementation**. We compare the performance of LLMLIFT against MetaLift [25][4]. MetaLift uses a symbolic solver (Rosette [26]) to perform the search. We evaluate on the same 45 benchmarks as MetaLift. All the benchmarks have loops and require loop invariants to prove the functional equivalence of the source and the generated program. MetaLift solves 40 out of 45 with a timeout of 1 hour. LLMLIFT is able to solve **44**, i.e., generate the correct translation as well as the required invariants to prove the correctness. LLMLIFT solves 4 additional benchmarks on which MetaLift times out. In addition to solving more benchmarks, LLMLIFT solves them much faster. It takes less than **1** minute on average to solve each benchmark when MetaLift has to take an average of 3 minutes to solve. The amount of effort required to build LLMLIFT is also significantly less than MetaLift as it does not require the developers to provide any search-space description for $PS$ and invariants. As MetaLift requires over 1000 LoC for the description of these search-space, LLMLIFT requires only $\approx$100 lines of prompt.

## 4.2 Network Packet Processing

Network packet processing hardware, such as routers and switches, lacks flexibility post-development, preventing experimentation with new data-plane algorithms. Recently, a verified lifting approach [22] was introduced to simplify this process. This compiler offers the developers with two constructs: 1. a packet transaction language (subset of the C language) to express the semantics of these data-plane algorithms 2. a compiler [22] that translates the packet processing algorithms to the instruction set of programmable switch devices. Atoms are introduced as an instruction set of the hardware to represent the atomic operations supported by the hardware. Compiler translates the packet transaction algorithm to a sequence of atoms resulting in a different programmable switch configuration.

**LLMLIFT implementation**. We implement the Domino compiler using LLMLIFT by defining the semantics of the atoms in the prompt. We compare the performance of our implementation against MetaLift's implementation. All benchmarks in Domino are imperative C programs without any loop constructs, so no loop invariants are required for these benchmarks. The generated $PS$ are verified using a SMT solver. MetaLift solves all the 10 benchmarks with an average time of 6 seconds. LLMLIFT is also able to transpile all the **10** benchmarks but with an average time of only **2** seconds. Similar to the Spark case study, we do not require developers to specify the search-space for $PS$. While MetaLift requires over $\approx$1100 LoC to describe this search-space, LLMLIFT only uses $\approx$70 lines of prompt. In summary, LLMLIFT shows similar performance to the existing compiler but can be built using much less effort.

## 4.3 TACO

Tensors form the key construct in machine learning and tensor compilers play an important role in optimizing these operations. TACO [23] is one such compiler which can automatically generate highly optimized code tailored to CPUs and GPUs. TACO's language represents the operations in a concise einsum like notation. Recently, C2TACO [6] a search-based lifting tool was proposed to automate the translation of C++ code to TACO.

**LLMLIFT implementation**. In Tab. 1, we compare the performance of C2TACO and LLMLIFT for all the benchmarks. We use the same 90 mins timeout for each benchmark that was used in the original C2TACO evaluation [6]. C2TACO solves 57 out of the total 60 benchmarks, while LLMLIFT successfully solves all **60** benchmarks. The 3 benchmarks that C2TACO fails to solve

---

[4]Casper [5] is not functional and Mold [1] is not open-sourced.

| Tool | BLAS | DSP | DSPStone | makespeare | mathfu | simpl_array | UTDSP | darknet |
|------|------|-----|----------|------------|--------|-------------|-------|---------|
| C2TACO | 100% | 100% | 100% | 100% | 91.6% | 90% | 100% | 92.8% |
| LLMLift | 100% | 100% | 100% | 100% | **100%** | 100% | **100%** | **100%** |

Table 1: Accuracy on various benchmarks for TACO.

require expressions of depth greater than 4. Due to its enumerative approach, C2TACO struggles to find solutions for these cases. We attempted to run these 3 challenging benchmarks with an extended timeout of 1 day, but the C2TACO was still unable to find a solution. C2TACO uses over 1000 LoC for implementing the heuristics to scale the symbolic search. In contrast, LLMLift relies on a simple **100** lines of prompt (task instruction + DSL semantics) to achieve better performance than C2TACO. C2TACO takes an average of 41 seconds while LLMLift average solving time is **2** seconds. We also perform an experiment to test the scalability of C2TACO enumerate apporach with more complex benchmarks than the ones used in the original evaluation. We include the results in Appendix C.

## 4.4 Tensor Processing

Many domains, such as image processing, signal processing, and deep learning, have legacy code written in high-level languages that operate on individual values of the input and perform specific operations. To leverage the optimizations provided by deep learning frameworks or hardware backends like GPUs, this code needs to be lifted to the operators supported by these languages. Prior work [24] introduced a tensor IR that can translate sequential programs to six different hardware and software backends automatically using a verified lifting approach.

**LLMLift implementation**. We evaluate LLMLift against Tenspiler [24] on the 23 benchmarks from the image processing and ML kernel domain.[5] Tenspiler is able to solve all 23 benchmarks. LLMLift also successfully solves all **23** benchmarks, including generation of the correct proofs.[6] However, it is important to note that Tenspiler's synthesis algorithm relies on three domain-specific optimizations to achieve scalability. These optimizations require significant effort to implement, with over $\approx$1200 LoC written by a domain expert. In contrast, LLMLift uses $\approx$320 lines of prompt to solve these benchmarks. It does not rely on any user-defined heuristics, which showcases its ability to generate correct solutions without the need for domain-specific optimizations. To check the scalability of Tenspiler's symblioc approach, we remove all the optimizations. Tenspiler, without the optimizations, can only solve 5 out of the 23 benchmarks with a timeout of 1 hour, highlighting the importance of the domain-specific optimizations for its performance. These results highlight the ability of LLMLift to solve complex benchmarks without relying on domain-specific heuristics. Moreover, LLMLift solves these benchmarks faster than Tenspiler with all its optimizations enabled. LLMLift takes an average time of **95.89** seconds to solve each benchmark, whereas Tenspiler takes 115.14 seconds.

## 4.5 Two-phase Approach for LLMLift

In this section, we evaluate an alternative approach to the two-phase method described in Sec. 3, where we generate the $Inv$(s) and the $PS$ together in a single step. To test this, we prompt the model in a one-shot setting, providing an example that demonstrates generating the $PS$ and the $Inv$(s) simultaneously. We merge the prompts described in Fig. 6 and Fig. 7 to create a unified prompt for this experiment.

Due to budget constraints, we limit this experiment to the tensor processing domain, which represents our most complex DSL with 37 operators. We use the same query budget as the two-phase approach. When prompted to generate the invariant and $PS$ together, LLMLift successfully solves 20 out of the total 23 benchmarks. In contrast, the two-phase approach described in Sec. 3 solves all **23** benchmarks. We hypothesize that the reduced performance of the single-phase approach may be attributed to the increased complexity of generating both the $PS$ and the $Inv$(s) simultaneously. Moreover, the two-phase approach enables the model to leverage the generated $PS$ when constructing the invariant. By having access to the $PS$, the model can more effectively reason about the necessary conditions and constraints required for the invariant to hold.

---

[5]We refer the readers to the paper [24] for more details on these benchmarks.

[6]We also evaluate LLMLift on these benchmarks using other LLMs and present the results in Appendix I.

## 5 Related Work

**Code Transpilation for DSLs.** Several approaches have been proposed for automating the task of translating legacy or unoptimized code to DSLs. These range from symbolic rule-based approaches [1] to search-based verified lifting approaches [7, 5, 27, 8, 25, 6] and neural approaches [9, 28]. Most of these tools are either optimized for a specific domain or require domain expertise to scale. In contrast, LLMLIFT simplifies the process of building lifting tools by leveraging LLMs. Closely related to our work is [29], where an IR is designed for low-resource languages and then combination of LLM and compiler techniques is used to reliably generate code for these languages.

**Code Translation for Mainstream-to-Mainstream Languages**. The closest work to ours is by Roziere et al. [28] on a sequence-to-sequence model to translate code between C++, Java, and Python. Our work differs in two key respects: we target lifting to DSLs, and our LLM based approach produces formally verified code. The objective of verified lifting is to map functional programs to the operators of a DSL in a semantics-preserving manner. Translating between mainstream languages has its own set of challenges: 1. Different languages support various constructs, making direct mapping challenging; 2. Disparities in type systems across languages must be handled, and, 3. Generating accurate verification conditions and formal proofs for equivalence checking across diverse language constructs is complex. Due to these challenges, prior work in mainstream-to-mainstream translation, such as [28], often relies on test-case-based approaches to demonstrate semantic equivalence, rather than formal verification. Such approaches do not provide any correctness guarantee in the generated code, and hence are risky to deploy in practice.

**LLMs for Code.** LLMs are trained on massive amounts of code from various sources, leading to impressive performance on programming tasks such as code generation [12, 13], repair, testing, and transpilation. While the use of learning to synthesize proof artifacts, specifications, and models in formal methods is not new [18], recently LLMs have been successfully employed in such tasks (e.g., [15, 16, 29]). However, generating reliable code from LLMs remains challenging due to the stochastic nature of these models and the difficulty in creating a verification oracle for complex specifications. With LLMLIFT, we demonstrate the first approach to verified code generation with LLMs, albeit in the limited setting of transpilation for side-effect-free code.

## 6 Conclusion

We presented a principled approach to leverage LLMs for code transpilation. Unlike prior LLM-based transpilers, our transpiled code is *provably equivalent* to the input, while also takes significantly less time to generate as compared to prior non LLM-based approaches with correctness guarantees, as demonstrated in transpiling to 4 DSLs used across a range of application domains.

## 7 Limitations

While LLMLIFT demonstrates impressive performance across four DSLs, there are few opportunities for future improvements. Currently, our approach generates a program summary and checks only for syntactic correctness, ensuring that the generated expressions are compatible with the SMT solver. We then generate invariants corresponding to the program summary, which are formally verified for correctness. However, incorporating a semantic filtering step using test cases could potentially eliminate some spurious program summaries. Another limitation of our current approach is that we use Boolean feedback to check the correctness of a solution. Providing more granular feedback, such as counter-examples from the theorem prover or compiler error messages, can possibly help guide the LLM towards generating correct solutions more efficiently.

## Acknowledgments and Disclosure of Funding

We would like to thank Federico Mora Rocha, Elizabeth Polgreen, Rishabh Singh and the anonymous reviewers for their insightful feedback.

This work was supported in part by DARPA Contract FA8750-23-C-0080 (ANSR), a Google BAIR Commons project, C3DTI, NSF grants IIS-1955488, IIS-2027575, ARO W911NF2110339, ONR N00014-21-1-2724, and DOE award DE-SC0016260, DE-SC0021982, and the Sloan Foundation.

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

# Appendix

## A   Algorithm

The `transpile_code` algorithm shown in Fig. 5 translates the source code of a program into a target language using LLM. The `source_code` parameter represents the code to be transpiled, and `num_iters` and n specify the number of rounds to generate $PS$ and $Inv$s and the number of $PS$ and $Inv$s to generate in each round, respectively.

The `seen_ps_sols` set keeps track of all $PS$ that have been processed by the algorithm. The algorithm operates in a loop that runs for `num_iters` iterations (line 4). In each iteration, the algorithm calls `get_ps_sols` (line 8) to generate n different $PS$ solutions for the given `source_code` from LLM, and tells LLM to not generate any $PS$ in the `seen_ps_sols` set. This is because $PS$ solutions are processed in previous iterations and did not yield verifiable translations, so we deem them incorrect. For each generated $PS$, the algorithm first checks if it has been seen before by looking it up in the `seen_ps_sols` set (line 11). If the $PS$ has been encountered previously, the algorithm skips it to avoid redundant processing. If it is new, the algorithm parses it to check for syntactic validity using the parse function (line 19). If the summary has invalid syntax, it is discarded, and the algorithm moves on to the next summary. If the $PS$ is syntactically correct, the algorithm proceeds to generate $Inv$(s) for it. It maintains a set called `seen_inv_sols_for_ps` that keeps track of invariants that have been processed for the current $PS$ to avoid redundant processing. It also checks each generated $Inv$(s) syntactic validity and discards it if it is not.

If both the $PS$ and the $Inv$(s) pass the syntactic validation, the algorithm proceeds to verify their correctness using the `verify` function (line 44). If the verification succeeds, the algorithm returns the $PS$ (`ps_sol`) as the final transpiled code (line 45).

The algorithm continues this process of generating $PS$ and $Inv$s and verifying their correctness until a valid solution is found or the maximum number of tries is reached. If no valid solution is found within the given number of tries, the algorithm returns `None` (line 48), indicating that the transpilation was unsuccessful.

## B   Prompts

In this section, we present an instantiation of the prompt structure shown in Fig. 4. The prompt shown in Fig. 6 consists of several components designed to guide the language model in generating semantically equivalent code using a restricted set of functions and constants. The prompt begins with a clear task description that instructs the model that its goal is to rewrite the given C++ function using only the provided functions and constants while maintaining semantic equivalence. Next, the prompt includes a set of instructions. These constraints are designed to make the generated code easier to parse and translate into a format suitable for theorem provers. The prompt then provides a set of defined functions in Python. These functions define all DSL operators that the model can use to rewrite the given C++ function. Finally, the prompt includes the test function in C++ which the model should rewrite using the provided functions and constants.

In Fig. 7, we present a one-shot prompt designed to guide the language model in generating loop invariants for the given `test` function. This prompt is similar in structure to the program summary guessing prompt: it provides a clear task description, a set of instructions, and examples to guide the model in generating the desired output. The prompt instructs the model to prove the assertion in the `test` function by finding a loop invariant using the defined functions. It includes specific constraints on the generated loop invariant, such as using only the defined functions, avoiding loops, using a single return statement, inlining expressions, and generating separate invariants for each loop in the `test` function. These constraints are intended to simplify the parsing of the generated invariants into SMT formulas, making it easier to integrate them into automated theorem provers. Additionally, the prompt provides a template for the invariant structure, guiding the model in constructing the loop invariant as a Python function that takes the loop variables and relevant data structures as input and returns a boolean expression. The invariant should involve comparisons of the loop variables using operators and expressions, and an equality check for the loop-dependent variable using the defined functions. The prompt also includes an example to demonstrate the expected format and structure of the loop invariant. Example 1 shows a `test` function that performs element-wise subtraction of two

```
1   def transpile_code(source_code: str, num_iters: int, n: int) -> str:
2     seen_ps_sols: set[str] = set()
3
4     for _ in range(num_iters):
5       # Gets `n` PS solutions from LLM for the given `source_code`.
6       # Assuming all solutions in `seen_ps_sols` are incorrect, `get_ps_sols`
7       # tells the LLM to not generate any solution in this set.
8       ps_sols: list[str] = get_ps_sols(n, source_code, seen_ps_sols)
9
10      for ps_sol in ps_sols:
11        if ps_sol in seen_ps_sols:
12          # We have processed this PS before.
13          continue
14        else:
15          # Otherwise, we add it to the set of seen PS solutions.
16          seen_ps_sols.add(ps_sol)
17
18        # If this PS has invalid syntax
19        if not parse(ps_sol):
20          continue
21
22        # Generate invariants for this PS
23        seen_inv_sols_for_ps: set[str] = set()
24        # Gets `n` Inv solutions from LLM for the given PS solution `ps_sol`.
25        inv_sols: list[str] = get_inv_sols_for_ps(n, ps_sol)
26        for inv_sol in inv_sols:
27          if inv_sol in seen_inv_sols_for_ps:
28            # We have processed this Inv solution for this PS before.
29            continue
30          else:
31            # Otherwise, we add it to the set of seen Inv solutions for this PS.
32            seen_inv_sols_for_ps.add(inv_sol)
33
34          # If this Inv has invalid syntax.
35          if not parse(inv_sol):
36            continue
37
38          # Verify Inv and PS. In this call, we convert all PS and Inv solutions
39          # to the syntax of an SMT solver. The SMT solver then checks for semantic
40          # equivalence between the PS solution and the source program, using the
41          # generated invariants as proofs.
42          # `verify` returns True if and only if the
43          # PS solution is equivalent to the source program.
44          if verify(inv_sol, ps_sol):
45            return ps_sol
46
47    # No solution has been found.
48    return None
```

Figure 5: LLMLɪꜰᴛ's algorithm for generating $PS$ and $Inv$s.

matrices and provides two loop invariants. Example 2 shows the `test` function for which we need to generate the loop invariants.

To avoid regenerating the same incorrect solutions, the prompt also includes all the syntactically correct solutions that have been generated so far, along with a message saying *These generated programs are incorrect. Do not generate the same. Please generate another program.*

---

**PS Guessing Prompt**

Your task is to rewrite the given test C++ Function. You need to use only the set of provided functions and constants to achieve this. The rewritten program should be semantically equivalent to the test function. Please do not generate any explanations.
#Instructions
# 1. Do not use for/while loops for rewriting the function.
# 2. The rewritten program should just be a single return statement of the form return_var = provided_function(...)
# 3. Inline all the expressions. Do not use intermediate variables.
```
#defined functions
from typing import Any, Callable, List

def ite(cond: bool, if_then: Any, if_else: Any) -> Any:
  return if_then if cond else if_else

def reduce_sum(x: List[int]) -> int:
```

```python
    return 0 if len(x) < 1 else x[0] + reduce_sum(x[1:])

def vec_elemwise_mul(x: List[int], y: List[int]) -> List[int]:
    return (
        []
        if len(x) < 1 or not len(x) == len(y)
        else [x[0] * y[0], *vec_elemwise_mul(x[1:], y[1:])]
    )

def reduce_max(x: List[int]) -> int:
    return (
        x[0]
        if len(x) <= 1
        else (x[0] if x[0] > reduce_max(x[1:]) else reduce_max(x[1:]))
    )

def vec_elemwise_add(x: List[int], y: List[int]) -> List[int]:
    return (
        []
        if len(x) < 1 or not len(x) == len(y)
        else [x[0] + y[0], *vec_elemwise_add(x[1:], y[1:])]
    )

def vec_elemwise_sub(x: List[int], y: List[int]) -> List[int]:
    return (
        []
        if len(x) < 1 or not len(x) == len(y)
        else [x[0] - y[0], *vec_elemwise_sub(x[1:], y[1:])]
    )

def vec_elemwise_div(x: List[int], y: List[int]) -> List[int]:
    return (
        []
        if len(x) < 1 or not len(x) == len(y)
        else [x[0] // y[0], *vec_elemwise_div(x[1:], y[1:])]
    )

...

//test function
#include <vector>
using namespace std;

int test(vector<int> input, int max_pos) {
    int max_val = input[0];
    for (int i = 1; i < max_pos; i++)
        if (input[i] > max_val)
            max_val = input[i];
    return max_val;
}
```

Figure 6: Prompt for guessing the $PS$.

## Invariant Guessing Prompt

Your task is to prove that 'assertion' is true in the 'test' function. The assertion can be proved by finding a loop invariant using the defined functions. Write the loop invariant as a python boolean formula.
#Instructions:
1. You need to use only the defined functions to write the loop invariant.
2. Do not use for/while loops for rewriting the function.
3. The rewritten program should just be a single return statement of the form return_var = provided_function(...)
4. Inline all the expressions. Do not use intermediate variables.
5. Generate separate loop invariants for each loop in the test function.
6. invariant structure
def invariant(i: int, input: List[int], ss: int, weight: List[int]) -> bool: return i op expr() and i op expr() and ss == operation over defined functions

Example1:
```python
#defined functions
def vec_elemwise_sub(x: list[int], y: list[int]) -> list[int]:
  return (
    []
    if len(x) < 1 or not len(x) == len(y)
    else [x[0] - y[0], *vec_elemwise_sub(x[1:], y[1:])]
  )
def matrix_elemwise_sub(matrix_x,: list[list[int]], matrix_y: list[list[int]]) -> list[list[int]]:
  return (
    []
    if len(matrix_x) < 1 or not len(matrix_x) == len(matrix_y)
    else [
      vec_elemwise_sub(matrix_x[0], matrix_y[0]),
      *matrix_elemwise_sub(matrix_x[1:], matrix_y[1:]),
    ]
  )
```
```cpp
//test function
vector<vector<uint8_t>> test(vector<vector<uint8_t>> base, vector<vector<uint8_t>> active) {
  vector<vector<uint8_t>> out;
  uint8_t m = base.size();
  uint8_t n = base[0].size();
  for (uint8_t row = 0; row < m; row++) {
    vector<uint8_t> row_vec;
    for (uint8_t col = 0; col < n; col++) {
    uint8_t pixel = base[row][col] - active[row][col] ;
    row_vec.push_back(pixel);

    }
    out.push_back(row_vec);
  }
  assert out == matrix_elemwise_sub(base, active);
}
```
```python
def invariant1(row, col, base, active, out):
  return row >= 0 and row <= base.size() and out == matrix_elemwise_sub(base[:row], active[:row])
def invariant2(row, col, base, active, row_vec, out):
  return row >= 0 and row < base.size() and col >= 0 and col <= base[0].size() and
    row_vec == vec_elemwise_sub(base[row][:col], active[row][:col]) and
    out == matrix_elemwise_sub(base[:row], active[:row])
```

Example2:
```python
#defined functions
from typing import Callable, List
def matrix_scalar_sub(a: int, matrix_x: List[List[int]]) -> List[List[int]]:
  return (
    []
    if len(matrix_x) < 1
    else [vec_scalar_sub(a, matrix_x[0]), *matrix_scalar_sub(a, matrix_x[1:])]
  )

def matrix_scalar_mul(a: int, matrix_x: List[List[int]]) -> List[List[int]]:
  return (
    []
    if len(matrix_x) < 1
    else [vec_scalar_mul(a, matrix_x[0]), *matrix_scalar_mul(a, matrix_x[1:])]
  )

def matrix_scalar_div(a: int, matrix_x: List[List[int]]) -> List[List[int]]:
  return (
    []
    if len(matrix_x) < 1
    else [vec_scalar_div(a, matrix_x[0]), *matrix_scalar_div(a, matrix_x[1:])]
  )
```

```
def scalar_matrix_sub(a: int, matrix_x: List[List[int]]) -> List[List[int]]:
  return (
    []
    if len(matrix_x) < 1
    else [scalar_vec_sub(a, matrix_x[0]), *scalar_matrix_sub(a, matrix_x[1:])]
  )

...

//test function
#include <vector>
using namespace std;

int rmsnorm_part1(vector<int> input, vector<int> weight) {
  int ss = 0;
  for (int i = 0; i < input.size(); i++)
    ss += input[i] * input[i];
  assert ss == reduce_sum(vec_elemwise_mul(input, input))
}
```

Figure 7: Prompt for guessing the loop invariant(s).

## C  Scalability

```
1   void fourth_in_place(int* arr, int n) {
2     for (int i = 0; i < n; ++i) {
3       arr[i] = arr[i] * arr[i];
4       arr[i] = arr[i] * arr[i];
5     }
6   }
7   //TACO expression
8   out[i] = arr[i] * arr[i] * arr[i] * arr[i]
9
10  //Incorrect TACO expressions
11  out(i) = arr(i) * arr(i)
12  out(i) = Cons1 + arr(j,i)
13  out(k) = Cons1 * arr(l,k,j)
```

```
1   void test1(int* arr, int n) {
2     for (int i = 0; i < n; ++i) {
3       arr[i] = arr[i] + arr[i] + arr[i] + arr[i] + arr[i];
4     }
5   }
6   //TACO expression
7   out[i] = arr[i] + arr[i] + arr[i] + arr[i] + arr[i]
```

(a) Benchmark on which C2TACO fails.

(b) Example of synthetic benchmark with expression length $= 5$.

Figure 8: Scalability evaluation of symbolic solvers using synthetic benchmarks.

In this section, we evaluate the scalability of symbolic solvers in the context of verified lifting-based tools. The benchmarks used in the evaluation of these tools are often carefully selected and limited in scope, allowing the tools to perform well within their intended domain. However, in this experiment, we aim to demonstrate that symbolic tools relying on domain-specific heuristics can be brittle and fail to scale when the complexity of the benchmarks increases beyond a certain threshold.

We begin by evaluating C2TACO. Upon careful analysis of the benchmarks on which C2TACO struggles, we observed that the tool often times out when tasked with generating expressions of length greater than 4 One such example is illustrated in Fig. 8a where the source performs an in-place operation on an array arr of length n and raises each element of the array to the power of 4. C2TACO enumerates candidate expressions using tensor operators and index variables in increasing order of expression length. C2TACO enumerates ≈**30k** candidates. We illustrate some of the incorrect expressions in Fig. 8a.

To test the scalability of C2TACO, we randomly generated a set of 10 benchmarks with expressions of varying lengths, ranging from 5 to 10, incorporating various arithmetic operations (see Fig. 8b for an example). We used a timeout of 90 minutes for C2TACO, as reported in the original evaluation. C2TACO was unable to solve any of the 10 benchmarks within the timeout. In contrast, LLMLIFT, was able to solve all **10** benchmarks correctly in less than **2 seconds**. This performance can be attributed to the ability of language models to identify patterns and learn from the context provided in the source code. To further test the capabilities of LLMLIFT, we evaluated it on a variation of the benchmark shown in Fig. 8a, where each element of the array is raised to the power of 20 instead of

```
1   vector<vector<int>> screen_blend_8(vector<vector<int>> base, vector<vector<int>> active) {
2     vector<vector<int>> out;
3     int m = base.size();
4     int n = base[0].size();
5     for (int row = 0; row < m; row++) {
6       vector<int> row_vec;
7       for (int col = 0; col < n; col++) {
8         int pixel = base[row][col] + active[row][col] - (base[row][col] * active[row][col]) / 255;
9         row_vec.push_back(pixel);
10      }
11      out.push_back(row_vec);
12    }
13    return out;
14  }
```

Figure 9: screen_blend benchmark source code.

4. Despite the increased complexity of the expression, LLMLIFT was able to generate the correct solution efficiently.

# D    Qualitative Analysis of the Errors

In this section, we provide a qualitative analysis of the mistakes made by LLMs while generating code and proofs. In LLMLIFT, we use Python as the IR and the $PS$ and $Inv$(s) are generated in Python. The errors encountered can be classified into two categories: syntactic and semantic.

Syntactic errors occur when the generated code constructs are not compatible with the theorem prover. To mitigate this issue, we use a syntactic parser that translates the generated solutions to the language supported by the theorem prover. The parser ensures that only supported constructs are present in the solutions and rejects any candidates that do not comply with the theorem prover's syntax.

One common source of syntactic errors is the use of Python-specific constructs that are not supported by SMT solvers. Although we prompt the model to generate solutions using only the constructs provided in the prompt's scope, controlling the exact code generated by the model can be challenging. Fig. 10 illustrates examples of program summaries generated by GPT-4 for the screen blend benchmark that contain unsupported constructs. For instance, the first solution in Fig. 10 uses a for loop, which is not supported by SMT solvers. Similarly, the second and third solutions utilize Python's list comprehension syntax, which is also not directly supported by SMT solvers. List comprehension are supported in SMT solvers using empty lists and append functions, such as append(1, []).

Semantic errors occur when the generated code is syntactically correct but is semantically not equivalent to the given S. In the context of the screen blend benchmark (shown in Fig. 9), Fig. 11 illustrates two examples of semantically incorrect programs generated by GPT-4. The first program incorrectly subtracts a term from the base matrix instead of subtracting it from the sum of base and active matrices. The second program suffers from a similar issue. It subtracts an incorrect term from the active matrix. Specifically, the term being subtracted is matrix_elemwise_div(matrix_elemwise_mul(base, active), matrix_scalar_mul(32, matrix_elemwise_mul(base, active))), which is different from the one in the given program.

# E    Proof of Equivalence

We use Floyd-Hoare Logic (FHL) to establish the validity of generated programs [30]. In FHL, verification problem is represented as a Hoare triple $\{A\}P\{B\}$, where: 1. $A$ is the pre-condition, 2. $P$ is the program to be executed, and 3. $B$ is the post-condition. An example of a valid Hoare Triple is $\{x = 0\}\ x := x + 1\{x > 0\}$. The Hoare triple states that if $x = 0$ before executing $x := x + 1$, then after executing the program, $x$ will be greater than $0$. To establish the validity of a Hoare triple, we prove that all executions starting from states satisfying $A$, after executing program $P$, result in states satisfying $B$. This involves finding a Boolean predicate called the verification condition ($VC$) that characterizes the set of pre-conditions from which every execution of $P$ leads to a state satisfying $B$. Formally, we need to prove that the $VC$ is true given pre-condition, i.e., $A \rightarrow VC(P, B)$.

```
1  def screen_blend_8(base: List[List[int]], active: List[List[int]]) -> List[List[int]]:
2    return matrix_elemwise_add(
3      matrix_elemwise_sub(
4        base,
5        matrix_elemwise_div(
6          matrix_elemwise_mul(base, active),
7          vec_scalar_mul(32, [1 for _ in range(len(base[0]))])
8        )
9      ),
10     active
11   )
```

```
1  def screen_blend_8(base: List[List[int]], active: List[List[int]]) -> List[List[int]]:
2    return matrix_elemwise_add(
3        matrix_elemwise_sub(
4            base,
5            matrix_elemwise_div(
6                matrix_elemwise_mul(base, active),
7                vec_scalar_mul(32, [[1]*len(base[0])]*len(base))
8            )
9        ),
10       active
11   )
```

```
1  def screen_blend_8(base: List[List[int]], active: List[List[int]]) -> List[List[int]]:
2    return matrix_elemwise_add(
3      matrix_elemwise_sub(
4        base,
5        matrix_elemwise_div(
6          matrix_elemwise_mul(base, active),
7          vec_scalar_mul(32, [[1]*len(base[0])]*len(base))
8        )
9      ),
10     active
11   )
```

Figure 10: Programs rejected by LLMLIFT's syntactic parser.

```
1  def screen_blend_8(base: List[List[int]], active: List[List[int]]) -> List[List[int]]:
2    return matrix_elemwise_add(
3      matrix_elemwise_sub(
4        base,
5        matrix_elemwise_div(
6          matrix_elemwise_mul(base, active),
7          scalar_matrix_div(32, base)
8        )
9      ),
10     active
11   )
```

```
1  def screen_blend_8(base: List[List[int]], active: List[List[int]]) -> List[List[int]]:
2    return matrix_elemwise_add(
3      base,
4      matrix_elemwise_sub(
5        active,
6        matrix_elemwise_div(
7          matrix_elemwise_mul(base, active),
8          matrix_scalar_mul(32, matrix_elemwise_mul(base, active))
9        )
10     )
11   )
```

Figure 11: Programs rejected by theorem prover for semantic incorrectness.

Standard techniques exist to generate verification conditions from a given source program [31]. For programs containing loops, an additional predicate called a loop invariant is required. This invariant helps prove that the post-condition remains valid regardless of the number of loop iterations. The inference rules provided by FHL can be encoded into a format that can be fed into automated theorem provers or SMT solvers. This encoding allows for the mechanical checking of any Hoare triple's validity. In Fig. 13, we show the $VCs$ generated for the source program in Fig. 12.

```
1   vector<int> test(vector<int> base, vector<int> active) {
2     vector<int> out;
3     for (int i = 0; i < base.size(); ++i)
4       out.push_back(active[i] + base[i]);
5     return out;
6   }
```

Figure 12: Source Code (S) for proof.

| Initial Condition | $Inv(i = 0, out = \{\}, active, base)$ |
|---|---|
| Preservation | $Inv(i, out, active, base) \wedge (i < base.size()) \rightarrow$ 
 $Inv((i + 1), out.push\_back(active[i] + base[i]), active, base)$ |
| Termination | $Inv(i, out, active, base) \wedge \neg (i < base.size()) \rightarrow PS(out, active, base)$ |

Figure 13: Verification conditions for the source code in Fig. 12.

```
def invariant(data, i):
  return i >= 0 and i <= base.size() and out = vec_elemwise_add(active[:i], base[:i])
def PS(out,active,base):
    return out == vec_elemwise_add(active, base)
```

Figure 14: $PS$ and $Inv$ for the source code in Fig. 12.

Proof:

1. **Initial Condition**: Before the loop executes, $i = 0$ and $out = [\,]$. The loop invariant expresses $out$ as the result of a vec_elemewise_add operator over the first i elements of $active$ and $base$. Since $i = 0$, the vec_elemewise_add operation is applied to an empty list, resulting in an empty list. Therefore, the invariant holds in the initial state.

2. **Preservation Condition**: The preservation condition ensures that the invariant holds throughout all iterations of the loop. This can be shown by induction. Assume the invariant holds at the i-th iteration. In the (i + 1)-th iteration, vec_elemewise_add would compute the element-wise sum for the first i + 1 elements of $base$ and $active$, while the source program would push $active[i + 1] + base[i + 1]$ to $out$, making $out$ equal to vec_elemewise_add(active[:i+1], base[:i+1]), i.e., the RHS for this condition is true.

3. **Termination Condition**: The termination condition requires that the invariant implies the post-condition. When the loop terminates, $i = base.size()$, and both the $PS$ and $Inv$ expressions for out will be identical, i.e., the post-condition is satisfied.

# F  Code Generation

Once we have the verified program summary in the IR, the code generation phase uses syntax-driven rules to translate IR programs into the target DSL's concrete syntax. Given an IR expression, the code generation function parses its operators and operands and translates each of them recursively. Below, we show a snippet of a code generation function that translates IR programs to PyTorch. In this function, each IR expression type is matched against a translation rule: variables are mapped to their names (lines 2-3), literals are mapped to their values (lines 4-5), and function calls are mapped to their corresponding PyTorch operators (lines 6-11).

For instance, the elemwise_add operator in IR translates torch.add in PyTorch, and elemwise_sub translates to torch.subtract. As a result, the IR expression elemwise_add(elemwise_sub(a, b)) becomes torch.add(torch.subtract(a, b)) in PyTorch.

Similarly, if we want to translate the code to another DSL, such as TensorFlow, we can simply change the specific operator rules to generate TensorFlow code. For example, the elemwise_add operator

in IR would be translated to `tf.add` (line 9), and the `elemwise_sub` operator would be translated to `tf.subtract` (line 11). Thus, the same IR expression `elemwise_add(elemwise_sub(a, b))` would become `tf.add(tf.subtract(a, b))` in TensorFlow.

```python
def codegen(expr: Expr):
  if isinstance(expr, Var):
    return expr.name()
  elif isinstance(expr, Lit):
    return expr.val()
  elif isinstance(expr, Call):
    f_name, args = expr.name(), expr.arguments()
    if f_name == "elemwise_add":
      return f"torch.add({codegen(args[0])}, {codegen(args[1])})"
    elif f_name == "elemwise_sub":
      return f"torch.subtract({codegen(args[0])}, {codegen(args[1])})"
        ...
```

Figure 15: Code for translating IR expression to concrete syntax of DSL (PyTorch).

## G   Performance of Generated Code

The primary objective of verified lifting is to generate semantically equivalent programs in the target DSL from the source. DSLs are inherently designed to offer domain-specific optimizations, and the performance gains observed post-translation are attributable to the implementation of operators within the DSL rather than the translation process itself.

In LLMLIFT, our aim was to replicate the existing VL-based compilers. We performed a manual verification of LLMLIFT's output against the corresponding symbolic tools, confirming output equivalence of the two tools. Given this equivalence, the performance gains reported by the original symbolic tools are directly applicable to LLMLIFT's translations. Performance numbers for some of the domains are following:

1. **Tensor Processing**: The objective in this domain is to generate tensor programs executable on tensor processing backends. Translations to this intermediate representation (IR) yield performance gains of $2.1\times$ (NumPy) and $167.71\times$ (PyTorch) compared to sequential C++ implementations when compiled with GCC -O3.

2. **Distributed Computing**: The generated Spark implementations achieved an average speed-up of $15.6\times$ compared to the sequential Java implementations. Additionally, when compared to manually written code by an expert, the generated outputs performed competitively. For more details on the user study, we refer the reader to the paper

3. **TACO**: The TACO compiler generates optimized GPU code. Translating programs to the TACO IR results in a performance gain of $24\times$ compared to sequential C++ programs when compiled with GCC -O3.

It is important to note that finding a program with optimal performance on the target backend would require performing the search phase with specific cost (objective) functions. While finding an equivalent program in the target DSL is already a challenging task, incorporating an optimization function into the search adds another layer of complexity. In addition, defining these cost functions is non-trivial in itself, as they must accurately capture the performance characteristics of the target backend. Currently, even without using cost functions, LLMLIFT is still able to generate performant code, as described earlier.

## H   Python as IR

Python is one of the most widely represented programming languages in the training data of these models. This ensures that the model can generate Python code reliably with minimal prompting, reducing the likelihood of hallucinations. Python's highly syntactic nature facilitates easier parsing. This is beneficial as 1. it allows for the development of syntax-driven parsers that can efficiently

translate the IR to the language supported by theorem provers and 2. It simplifies the process of translating the IR to the target DSL's concrete syntax.

Direct conversion to a DSL is challenging because many DSLs may not be well-represented in the model's training data (as illustrated in Fig. 3). In addition, it is also challenging to verify DSL programs using theorem provers as they do not provide support for these languages and it is not trivial to translate the DSL directly to the language supported by them. On the other hand, many theorem provers have existing tools or libraries for handling Python-like syntax, making the verification step of LLMLIFT more reliable.

# I   LLMLIFT's Performance with Other Large Models

To evaluate if our prompts generalize across models, we evaluated LLMLIFT's performance using GPT-4o[32], Claude 3.5 Sonnet[33], and Gemini-1.5-Pro[34]. We used the exact same setting as described in Sec. 4 (zero-shot for $PS$ and one-shot for $Inv$) for this evaluation. We evaluated on the 23 benchmarks from the tensor processing domain. GPT-4o and Claude 3.5 Sonnet were able to solve all the benchmarks while Gemini-1.5-Pro solved 21 benchmarks (failed to guess the correct $PS$ for the 2 unsolved benchmarks). The results show that our prompts are robust and generalize well across models, even without prompt engineering.

# J   LLMLIFT's Performance with Smaller Models

To explore whether LLMLIFT could leverage smaller open-source models or if larger models like Claude and GPT-4 were necessary, we evaluated LLMLIFT using two recent open-source models, Llama3 8B[35] and Mistral Nemo 12.2B[36], using Hugging Face's serverless inference API. We used the benchmarks from the tensor processing domain, which is the most complex among all the DSLs we evaluated. We use the same budget of queries and temperature settings.

Neither Llama3 and Mistral solved any of the benchmarks. Their solutions used Python constructs outside of the defined IR, which caused them to immediately fail our syntactic parser. Our results suggest that larger models can be more effectively used for the task of VL without requiring fine-tuning, for the following two reasons:

1. **Instruction Following**: Larger models are better at following the instructions to generate programs strictly with the defined DSL operators.
2. **Handling Long-Context**: With prompts that can grow in size to include all DSL operators and feedback, larger models manage long-context tasks more effectively.

# K   LLMLIFT's Performance With Zero-Shot Prompt

At the time of writing, GPT-4 was the most advanced model, and we used one-shot prompting, as shown in Fig. 7, to generate loop invariants. Recently, with further improvements in the models, we experimented with a simplified prompt to test whether zero-shot prompting could effectively infer invariants. Note that this prompt is zero-shot but includes some natural language instructions to guide the model on the structure of the loop invariant. We evaluated LLMLIFT's performance using GPT-4o[32], Claude 3.5 Sonnet[33], and Gemini-1.5-Pro[34]. We used the exact same prompts for generating $PS$, and the prompt described in Fig. 16 for inferring loop invariants. We used the same temperature settings and query budget for this experiment. We queried these models using their API endpoints. We evaluated on the 23 benchmarks from the tensor processing domain as this domain has the most complex DSL. In addition, all of them have loops and require loop invariants for proving equivalence of the translated programs with the source programs.

As shown in Tab. 2, we observe that GPT-4, the model used for all experiments in Sec. 4, does not perform well with a zero-shot invariant prompt. A one-shot prompt, shown in Fig. 7, was necessary for generating accurate guesses of loop invariants. However, more recent models like GPT-4o and Claude 3.5 Sonnet were able to solve all 23 benchmarks even under the zero-shot setting. We also observed that Claude used the least number of queries to generate the correct solutions. Gemini did not solve 3 out of 23 benchmarks (failed to generate correct $PS$ for 2 benchmarks and correct $Inv$ for 1).

| Model name | # of benchmarks solved |
|---|---|
| GPT-4 | 11 |
| GPT-4o | 23 |
| Claude 3.5 Sonnet | 23 |
| Gemini-1.5-Pro | 20 |

Table 2: Zero-shot performance of various proprietary models on tensor benchmarks.

**Invariant Guessing Zero-Shot Prompt**

Your task is to generate the loop invariant 'Inv' such that it is true at all the locations it is defined at. Generate only a single 'Inv' expression which holds at all the locations. The invariant needs to be generated using only the functions defined below. Write the loop invariant as a python boolean formula.
#Instructions:
1. You can use the defined functions to write the loop invariant. Do not use any for loops or any other python construct.
2. Generate separate loop invariants for each loop in the test function. Return the loop invariant as a single boolean expression. Only return the invariant and no other code in a code block.

```python
def ite(cond: bool, if_then: Any, if_else: Any) -> Any:
  return if_then if cond else if_else

def reduce_sum(x: List[int]) -> int:
  return 0 if len(x) < 1 else x[0] + reduce_sum(x[1:])

def vec_elemwise_mul(x: List[int], y: List[int]) -> List[int]:
  return (
    []
    if len(x) < 1 or not len(x) == len(y)
    else [x[0] * y[0], *vec_elemwise_mul(x[1:], y[1:])]
  )

def reduce_max(x: List[int]) -> int:
  return (
    x[0]
    if len(x) <= 1
    else (x[0] if x[0] > reduce_max(x[1:]) else reduce_max(x[1:]))
  )

def vec_elemwise_add(x: List[int], y: List[int]) -> List[int]:
  return (
    []
    if len(x) < 1 or not len(x) == len(y)
    else [x[0] + y[0], *vec_elemwise_add(x[1:], y[1:])]
  )

def vec_elemwise_sub(x: List[int], y: List[int]) -> List[int]:
  return (
    []
    if len(x) < 1 or not len(x) == len(y)
    else [x[0] - y[0], *vec_elemwise_sub(x[1:], y[1:])]
  )

def vec_elemwise_div(x: List[int], y: List[int]) -> List[int]:
  return (
    []
    if len(x) < 1 or not len(x) == len(y)
    else [x[0] // y[0], *vec_elemwise_div(x[1:], y[1:])]
  )

...

//test function
#include <vector>
using namespace std;

int rmsnorm_part1(vector<int> input, vector<int> weight) {
  int ss = 0;
  for (int i = 0; i < input.size(); i++)
    ss += input[i] * input[i];
  assert ss == reduce_sum(vec_elemwise_mul(input, input));
```

```
}

Use the following template to generate the loop invariant

# A strong loop invariant should have the following properties:
# 1. It should have boolean expressions over the loop index variable `i` to describe the valid range of `i`.
# 2. It should have an inductive expression describing the output variable `ss` using the defined functions.

def invariant(i: int, input: List[int], ss: int, weight: List[int]) -> bool:
  return expression over loop index variable i and ss == operation over defined functions
```

Figure 16: Zero-shot prompt for guessing the loop invariant.

