# OpenReview forum: "Verified Code Transpilation with LLMs"
_NeurIPS.cc/2024/Conference — NeurIPS 2024 poster_

### Official Review · Reviewer_9szn · 2024-06-27

**Soundness:** 2
**Presentation:** 2
**Contribution:** 3
**Rating:** 4
**Confidence:** 4

**Summary:**

This paper proposes an LLM-based approach (LLMLIFT) to building verified lifting tools. LLMLIFT leverages LLMs to generated both code and proof annotations together. For four real-world DSLs, LLMLIFT not only outperforms previous symbolic-based tools in both the number of benchmarks transpiled and transpilation time, but also requires significantly less effort to build.

**Strengths:**

For the important task of automatically code transpilation for DSLs, this paper domonstrate a novel approach to generating and verifying the generated code using LLMs. While using LLMs, the key idea of VL is combined: Python is used as IR, which solves the shortcomings of direct translation based on LLMs. The experimental results are improved compared with the existing methods.

**Weaknesses:**

Some statements are not clear and accurate:
1. An important contribution and innovation of this paper is that generating and verifying the generated code using LLMs. However, to be precise, LLMs just seems to generates loop invariants for validation and then combine them with validators.
2. Contributions should be distinguished from strengths.
The experiment does not seem to well illustrate the advantages of using LLMs to validate generated code as proposed in this paper. For example, to what extent did this LLMs-based validation phase improve the experimental results?

**Questions:**

1. Compared with traditional VL, does the LLMs-based VL proposed in this paper have any other advantages besides being better in finding IR expression sequences? For example, what are the specific advantages of generating loop invariants compared with traditional methods?
2. It is mentioned that the model uses PS to generate invariants, then whether it will lead to the error of the invariant due to the error of PS generation?
3. Did LLMLIFT make any adjustments after Boolean Feedback?
4. How to verify correctness if no loops are included in the program.

**Limitations:**

Are there DSLs that are hard to define with Python as IR?

---

> ### Author Rebuttal · Authors · 2024-08-07
>
> **LLMs just seems to generates loop invariants for validation and then combine them with validators. Compared with traditional VL, does the LLMs-based VL proposed in this paper have any other advantages besides being better in finding IR expression sequences?**
>
> To clarify, LLMLift does not directly use LLMs for code validation. Instead, LLMs are used to guess program summaries and loop invariants, which are then passed to a theorem prover to check the validity of the generated program and its proof.
>
> The challenge of finding a correct program summary or an equivalent program in a domain-specific language (DSL) is significant. For example, in our tensor processing domain, the search space for some problems can reach approximately 100,000 expressions for a 20 line input program. Traditional symbolic tools depend on domain-specific heuristics to manage this complexity, whereas LLMLift simplifies the entire search phase by using a straightforward structured prompt, as detailed in the paper.
>
> The program verification community has devoted substantial effort over the years to address the problem of automatically inferring loop invariants. Various approaches have been proposed, including template-based synthesis [1], example-driven methods [2], and machine learning techniques [3]. What sets LLMLift apart is that it bridges two traditionally separate lines of work—program synthesis and formal verification—allowing us to leverage the strengths of both. By integrating LLMs in our iterative framework, we are able to generate candidate loop invariants and program summaries more flexibly and efficiently than traditional methods. To the best of our knowledge, LLMLift is the first approach that can simultaneously solve *both* the translation and loop invariant generation problems while generalizing across multiple domains.
>
> This capability not only significantly reduces the human effort required to build these techniques but also provides formal guarantees on the output of the LLMs—a task that has traditionally been very challenging.
>
>
>
> **It is mentioned that the model uses PS to generate invariants, then whether it will lead to the error of the invariant due to the error of PS generation?**
>
> In LLMLift, there is a possibility that the PS guessing phase might result in an incorrect program summary, as the model initially validates only for syntactic correctness. Since the loop invariants are generated based on this potentially incorrect PS, the generated invariants could also be incorrect. However, these erroneous solutions are caught and rejected during the verification phase of LLMLift. In this phase, a theorem prover is used to check for the semantic equivalence of the generated PS and invariants with the original source program (as described in how we prove equivalence earlier). If the generated PS and its associated invariants do not semantically match the source program, they are discarded. We provide examples of programs rejected by the theorem prover in Figure 10.
>
> **Did LLMLIFT make any adjustments after Boolean Feedback?**
>
> No, we do not make any adjustments after the Boolean feedback. We just prompt the model with the feedback and the solution if the solution is incorrect.
>
> **How to verify correctness if no loops are included in the program.**
>
> See general response.
>
> ### References
> [1] Pranav et al. Learning invariants using decision trees and implication counterexamples
>
> [2] Saswat et al. Data-Driven Inference of Representation Invariants
>
> [3] Xujie et al. Learning loop invariants for program verification

---

### Official Review · Reviewer_ZcXM · 2024-07-08

**Soundness:** 2
**Presentation:** 2
**Contribution:** 3
**Rating:** 3
**Confidence:** 4

**Summary:**

This work proposes LLMLift that leverages large language models (LLMs) to perform program transpilation. It first uses a prompt to lift the source program into an intermediate representation of the operators in the target language, called a program summary. Then, it prompts the LLM to generate loop invariants if necessary. The generated program summary and loop invariants are used to verify the equivalence of the source program and the program summary. This process is repeated until a correct program summary is found or the computation budget is reached. Finally, the program summary is rewritten into code in the target language using simple rules. The evaluation was performed on four tasks: distributed computing, network packet processing, TACO, and tensor processing. The results show that LLMLift can correctly transpile more benchmarks, spend less time, and require less effort to develop.

**Strengths:**

Code transpilation is an interesting and important problem for both research and industry. The paper has demonstrated a nice combination of LLMs and verification to obtain provably correct transpiled code. The evaluation is performed on a diverse set of four scenarios. Moreover, I think the approach can have good potential even for translation between mainstream programming languages.

**Weaknesses:**

My main concern for this paper is poor writing, in both syntactic and semantic level. The writing needs significant improvement in general. I also have some other concerns about the evaluation.

### 1. Syntactic Writing Issues
I found the following syntactic issues:
- Typos: Line 135 program program, Line 175 Contarary. There might be other typos so I suggest the authors run spell checks.
- The number of arguments received by $\phi$ are different in Equations (1) and (2).
- It is visually difficult to separate the citations from normal text. I suggest to add a pair of parentheses like the ICML format.

### 2. Semantic Writing Issues
The general idea of the approach is easy-to-understand, the paper lacks many technical details:
- How do you verify the equivalence of the source program and the program summary? The paper provides hardly any details on this, but I think this is a non-trivial task.
- How do you produce the target program from the program summary? Again, the paper mentions that this can be done using simple rules without giving any technical details.
- How difficult are the four tasks considered in the evaluation? Providing some end-to-end examples would be helpful.
- Why do you use one-shot prompting for generating program summaries but zero-shot for generating loop invariants?

### 3. Concerns about Evaluation
- LLMLift’s benefit lies mostly in speed. However, it does not solve significantly more benchmarks than baselines. Why is it the case?
- The paper claims that LLMLift reduces development effort compared to prior rule-based approaches. I am not sure if this is a fair comparison, because the development of LLMs, such as GPT-4 used in the paper, requires significant development effort and computation resources.
- The paper targets only transpilation from mainstream languages to DSLs. What are the main obstacles and efforts needed to extend LLMLift to transpilation between mainstream languages?

**Questions:**

Please consider addressing the points raised in the “Weakness” section.

**Limitations:**

The paper has sufficiently addressed the points concerning limitations and societal impact.

---

> ### Author Rebuttal · Authors · 2024-08-07
>
> **How do you produce the target program from the program summary?**
>
> Once we have the verified program summary in the IR, the code generation phase uses syntax-driven rules to map IR operators to the concrete syntax of the target DSL. Below we show a snippet of a code generation function that translates a tensor processing IR to PyTorch.  This code generation function recursively parses the IR operators and maps them to the corresponding DSL operators.
>
> For instance, the `elemwise_add` operator in IR translates `torch.add` in PyTorch, and `elemwise_sub` translates to `torch.subtract`. Thus, the IR expression `elemwise_add(elemwise_sub(a, b))` becomes `torch.add(torch.subtract(a, b))` in PyTorch.
> ```python
> def codegen(expr: Expr):
>   if isinstance(expr, Var): return expr.name()
>   elif isinstance(expr, Lit): return expr.val()
>   elif isinstance(expr, Call):
>     f_name, args = expr.name(), expr.arguments()
>     if f_name == "elemwise_add":
>       return f"torch.add({codegen(args[0])}, {codegen(args[1])})"
>     elif f_name == "elemwise_sub":
>       return f"torch.subtract({codegen(args[0])}, {codegen(args[1])})"
>         ...
> ```
>
> **Why 0 shot for PS and 1 for invariants:**
>
> The task of generating program summaries is relatively easier compared to generating loop invariants. The primary instruction for this task is that the language model should combine operators from the given DSL without introducing any external operators. This constraint is easily expressible in natural language.
>
> Generating invariants, on the other hand, is a more complex task for the following reasons:
> 1) Specific Template: While LLMs might have encountered loop invariants in their training data, the invariants required for verified lifting often follow a specific template that may not be well-represented in the model's training set. Loop invariants in verified lifting must use a particular syntactic structure. As described in Equation 3 of our paper, they should contain expressions over loop variables and include inductive expressions using the output variable.
> 2) Language Specificity: The generated invariants are in Python instead of SMT-LIB. This specific format requirement makes it easier to implement a pattern-matching parser for translating the invariants to SMT-LIB later in the process.
>
> Describing these structural and syntactic restrictions accurately in natural language is challenging. Hence we use a one shot prompt for generating the invariants.
>
> **LLMLift does not solve significantly more benchmarks than baselines**
>
> We compare LLMLift against symbolic implementations that were evaluated on a small, diverse set of benchmarks designed to showcase the effectiveness of their specific approaches. As a result, these tools achieve high accuracy on these benchmarks. To demonstrate the scalability of our approach, we created a set of 10 synthetic benchmarks for C2TACO, which are far more complex than those in the original set of benchmarks (see Appendix C). Our experiments show that LLMLift easily scales to handle these more challenging benchmarks, solving all 10 benchmarks in under 2 seconds. In contrast, C2TACO is unable to solve any of the 10 synthetic benchmarks with a 90 minute timeout.
>
> **LLMLift reduces development effort compared to prior rule-based approaches**
>
> One of the key strengths of LLMLift is its ability to scale the search process for VL without relying on domain-specific heuristics. This significantly reduces the human effort required to build VL based compilers, making the technology more accessible across various domains. As detailed in our experimental section, symbolic search techniques heavily rely on domain-specific heuristics for scalability. For instance, C2TACO uses over 1000 lines of code solely to describe these heuristics, which are specific to the TACO IR and cannot be easily reused for other domains. Even with these heuristics, their search fails to scale to more complex benchmarks beyond those used for evaluation in the paper (see Appendix C). In contrast to domain-specific heuristics, the development effort for LLMs is a one-time investment that can be leveraged across multiple domains and tasks. Once trained, these models can be applied to various VL tasks with minimal additional development.
>
> **Main obstacles and efforts needed to extend LLMLift to transpilation between mainstream languages**
>
> The objective of VL is different from translating programs between mainstream languages. Our approach focuses on mapping functional programs to the operators of a DSL in a semantics-preserving manner. As described in the paper, this problem of mapping to DSL is important as a) DSLs often provide a more concise way of expressing the same program compared to imperative implementations, enhancing maintainability and b) DSLs typically offer domain-specific optimizations that can significantly improve performance. However, there are several challenges in this mapping problem. In addition to the search space being huge, LLMLift must not only find a correct translation but also generate a proof of equivalence. This adds another layer of complexity.
>
> Translating between mainstream languages has its own set of challenges:
> 1. Different languages support various constructs, making direct mapping challenging,
> 2. Generating accurate verification conditions and formal proofs for equivalence checking across diverse language constructs is complex.
>
> Due to these challenges, prior work in mainstream-to-mainstream translation, such as TransCoder[1], often relies on test-case-based approaches to demonstrate semantic equivalence, rather than formal verification. Such approaches do not provide any correctness guarantee in the generated code, and hence are risky to deploy in practice. On the other hand, LLMLift formally verifies the code which is generated and proves the semantic equivalence with the source program.
>
>
> ### References
> [1] Baptiste Roziere et al. Unsupervised translation of programming languages.

---

> > ### Comment · Reviewer_ZcXM · 2024-08-08
> > **Thanks for the rebuttal**
> >
> > Thanks for submitting the rebuttal. I have read it but I believe the paper needs a major revision to reach the bar of NeurIPS. Therefore, I will keep my score. Below I provide a list of suggestions that might be helpful for the next iteration of the paper:
> > - Improve writing and formalization.
> > - Include the verification method and the way of generating target programs from program summaries into the appendix.
> > - Design and present benchmarks that can better demonstrate LLMLift’s advantages, especially in solving more test cases.
> > - Include ablation studies on important hyperparameters.

---

### Official Review · Reviewer_KVm8 · 2024-07-11

**Soundness:** 3
**Presentation:** 3
**Contribution:** 3
**Rating:** 7
**Confidence:** 3

**Summary:**

The authors propose an approach named LLMLift, which utilizes large language models (LLMs) to achieve verified code transpilation. LLMLift not only translates a given program into its corresponding equivalent in the target language but also generates proofs for functional equivalence. The paper claims that this method outperforms previous symbolic-based tools in both the number of benchmarks transpiled and transpilation time, while requiring significantly less effort to build.

**Strengths:**

- The paper is highly innovative. It fully explores and leverages the generalization capability, few-shot learning capability, and inherent domain knowledge of LLMs. Additionally, it transforms the code translation task from end-to-end generation into generating PS and INV with LLMs, which cleverly reduces task difficulty and enhances the utilization of LLMs.

- The evaluation is thorough, covering multiple DSLs and comparison against a tensplier proposed in 2024, which fully demonstrates the capability of the method.

**Weaknesses:**

- The paper mentions a budget of 50 queries for the program synthesis (PS) and a budget of 10 queries for each PS, but it lacks an analysis of the success rate and the number of queries used. Providing this information would offer a clearer picture of the efficiency and effectiveness of the approach

**Questions:**

- Why is there no performance comparison given in the experimental evaluation? Is it because the task is merely to translate DSLs and is not related to performance issues, or does performance remain similar as long as the DSL is generated?
- How does the approach handle complex features in source languages that do not have direct counterparts in the target DSLs?
- What are the common factors in tests that failed, and how can the approach be adjusted to address these issues?
- Why use GPT-3.5 other than the latest version of GPT in Figure 3?

**Limitations:**

The authors have not adequately discussed the limitations of this paper.

---

> ### Author Rebuttal · Authors · 2024-08-07
>
> **Why use GPT-3.5?**
>
> The use of GPT-3.5 in Figure 3 was to illustrate a key challenge in applying LLMs to the task of verified lifting: the difficulty these models face in generating correct code for DSLs, even when the DSL is not particularly new. Our experiments with GPT-3.5 showed that despite TACO being introduced in 2017, which is well within the training cutoff date for GPT-3.5 (2021), the model still struggled to generate correct TACO code when used as-is.
> &emsp;This issue persists even with more recent and advanced models. For instance, when we tested the same prompt with GPT-4, which presumably has been trained on a larger corpus potentially including more TACO programs, it still produced an incorrect solution. Specifically, the generated program contained an incorrect usage of the get function in the final return statement.
> ```
> int mag_array(int *a, int n) {
>     Tensor<int> A({n}, Format{Dense}); Tensor<int> result(ScalarType::Int);
>     for (int i = 0; i < n; ++i) { A.insert({i}, a[i]); }
>     A.pack(); IndexVar i;
>     result() = sum(i, A(i) * A(i)); result.compute();
>     return result.getStorage().getValues().get<int>(0);
> }
> ```
> Similarly, when we tested Claude 3.5 Sonnet, one of the latest models from Anthropic, with the same prompt, it also returned incorrect code, misusing the sum function.
> ```
> int mag_array(int* a, int n) {
>     Format csr({Sparse, Dense}); Format dense({Dense});
>     Tensor<int> A({n}, csr); Tensor<int> B({n}, dense); Tensor<int> result({}, dense);
>     for (int i = 0; i < n; i++) { A.insert({i}, a[i]); }
>     A.pack(); IndexVar i; B(i) = A(i) * A(i);
>     result() = sum(B(i)); result.compile(); result.assemble(); result.compute();
>     return result.at({});
> }
> ```
> These results show that even the largest and most advanced LLMs struggle to reliably generate correct code for new DSLs when directly prompted to do so. This highlights the inherent difficulty of generating code reliably in new DSLs and illustrating that LLMLift is solving a challenging problem.
>
>
> **How does the approach handle complex features in source languages that do not have direct counterparts in the target DSLs?**
>
> LLMLift currently only supports a subset of the C/C++ and Python language in the source programs. In particular, it does not support any code that uses pointers or objects as verifying programs with these constructs is challenging. That said, we did not encounter the use of these constructs in any of the benchmarks in all the four domains that we evaluated on. Moreover, we currently support commonly-used data structures including arrays, tuples, maps, vectors and tensors, and that was enough for us to show good experimental results.
>
> **The paper mentions a budget of 50 queries for the program synthesis (PS) and a budget of 10 queries for each PS, but it lacks an analysis of the success rate and the number of queries used. Providing this information would offer a clearer picture of the efficiency and effectiveness of the approach.**
>
> The average number of tries to get to the correct PS solution is 4.3 and the median is 1, which means that most benchmarks generate the correct PS in the first try. Given a correct PS, it takes an average and a median of 1.77 and 1.74 tries, respectively, to generate the correct invariant.
>
> ### References
> [1] Maaz Bin Safeer Ahmad et al. Automatically leveraging mapreduce frameworks for data-intensive applications.

---

### Official Review · Reviewer_2Wxn · 2024-07-12

**Soundness:** 3
**Presentation:** 3
**Contribution:** 2
**Rating:** 3
**Confidence:** 3

**Summary:**

The paper aims to solve the problem of conversion of source code from one language to any domain specific language (DSL) leveraging LLMs eliminating manual intervention. The primary contribution lies in the domain of automating the transpilation along with providing functional correctness proof simultaneously. Their technique LLMLift converts source language to an intermediate representation (IR, python in this work), generates program summary and invariants, uses program summary to check the functional correctness using a verifier and finally converts to a DSL. They have evaluated multiple benchmarks across domains: distributed computing, network packet processing, tensor processing and TACO (tensor processing compiler for GPU code). Their evaluations set them apart by improving on latency (6x on average) and having high semantic accuracy.

**Strengths:**

1.	Timely problem considering the emergence of different accelerators.
2.	Automated transpilation with integrated feedback and verifier makes the process independent and self-contained.
3.	Use of python as an IR improves user readability and is more probable to be accurate given the huge online corpus and training data on the same.
4.	Zero-shot approach requires no retraining, but this could lead to issues as well. See weakness and detailed comments.

**Weaknesses:**

1.	DSLs have less representation in the present LLM training dataset (more so lacking correct and efficient codebases).
2.	Limited novelty, as there already exists work, and the authors confirm the same, which use LLMs to transpile from one language to another and produce proof annotations separately. The effort in the current work is largely automating and integrating these two steps.
3.	Underutilization of the GPT4 model which has a huge memory and compute footprint.
4.	The zero-shot approach might not yield very good results if the LLM does not have sufficient training data on the DSL. On the other hand, a solution like RAG is more feasible as it can be finetuned and made context aware with minimum information on the DSL.
5.	The end-to-end process of transpilation is not very clear. Although it sounds intuitive, the fine-grained steps are missing in the explanation, leading to a belief that the paper majorly contributes towards an efficient prompt-engineering for doing transpilation rather than having an innovative approach.
6.	 The evaluation metrics are poorly chosen. Using LoC as a metric of effort is not fair where the model and the framework (GPT4, and it’s API along with the implementation platform) doing the transpilation work has a resource (compute, memory, LoC, energy etc.) requirement of many orders of magnitude higher than any of the conventional tools like C2TACO.
7.	Although the evaluation consists of many benchmarks, we are missing comparisons for many important DSLs like HDLs, GPU programming, OpenCL etc. These languages form the foundation of most of the new generations of the domain specific accelerators design framework.

**Questions:**

See the part marked using (***)  below.

**Limitations:**

Leveraging generative models to solve the problem of transpilation and providing quantitative estimates of functional correctness simultaneously seems to be a timely and much needed solution in today’s tech diaspora. But usage of a model like GPT4 seems to be an overkill considering the resource requirement that it entails. Using a smaller model, that is trained to do sequence to sequence conversion along with target language library as a database in case of RAG can be thought of as a more efficient way to achieve the same. In the suggested method, there is dependency/ sole reliance on the model only to do the conversion to IR and then the target language, but not all DSLs would find the same degree of contribution in the training dataset of the models. Using RAG here gives the user more freedom.

The writing does not make it easy for the readers to understand the underlying sub-components and the handshake thereby. For instance, we do not know how the verifier works, the criteria on which it works and its logical complexities.

There seems to be a scalability issue for bigger codebases which would overshoot the supported prompt length of the model. We do not know how the code base is broken down into cohesive pieces and each forms a prompt. Also, for large code bases we see KV cache management, storage and communication to different compute nodes to be a serious problem.

(***) The paper does not include the rationale behind many of their assumptions like:
a.	Why python was chosen as the IR, and how much of a benefit (in terms of correctness/ latency) it provides compared to a direct conversion without an IR.
b.	Why is accuracy the right metric? Given we do not have much knowledge about the length, functionality and complexity of the source and the target languages. The authors fail to provide a comparison between the transpiled code and the DSL implementation designed by an expert.
c.	Why was a temperature of 0.7 chosen and what is the impact of varying the same?
d.	Why was GPT 4 specifically chosen given its huge size? And how would the design fair against self correcting models and AutoGPT kind of models?
e.	The paper overall misses sensitivity and ablation studies making it difficult, if not impossible, to understand the impact of each of the design components.

---

> ### Author Rebuttal · Authors · 2024-08-07
>
> **Why GPT-4**
>
> With LLMLift, our objective was to demonstrate that LLMs can be effectively used for the task of VL without requiring fine-tuning. We needed a model that was good in two things:
> 1. Instruction Following: The model needed to generate programs strictly using the defined operators in a DSL.
> 2. Handling Long-Context: The prompt can grow in size when incorporating all DSL operators and feedback, so the model needed to manage long-context tasks effectively.
> GPT-4 was the best model available that met both of these requirements.
>
> To explore whether LLMLift could leverage open-source models or if larger models were necessary, we evaluated LLMLift using two recent open-source models (Llama3 8B and Mistral Nemo 12.2B) and another proprietary model (Claude-sonnet-3.5). Due to budget constraints, we used a subset of benchmarks from the tensor processing domain, which is the most complex among all the DSLs we evaluated. Specifically, we used 12 image processing blend benchmarks. We use the same budget of queries and temperature settings.
>
> Open-source models, Llama3 and Mistral, did not solve any of the benchmarks. Their solutions use Python constructs outside of the defined IR, causing them to immediately fail our syntactic parser. Claude, on the other hand, successfully solved 9 out of the 12. Llama3 and Mistral are significantly smaller than proprietary models like Claude and GPT-4. The results suggest that larger models are better suited for VL.
>
> **Complexity of source programs**
>
> Lines of code (LOC) are often used as a proxy for evaluating program complexity. In the benchmarks, the average LOC is 14. As described in the paper, our aim is for LLMLift to translate functional programs, and we observe that 10-20 lines of code are typical for these programs in the real world.
>
> **RAG is more feasible solution**
>
> To clarify, LLMLift does not depend on the presence of specific DSLs in the training data of large language models to perform code translation. In fact, we demonstrate that LLMLift can reliably generate code in DSLs that the model may have never encountered during training, for instance our tensor processing DSLs and the TACO IR. With LLMLift, we introduce a structured prompting approach that explicitly includes the semantics of the DSL operators using an IR. The model's task is then to generate a program using only the operators defined in the prompt. Once we have a verified solution in the IR, we apply syntax-driven rules to translate the program into the concrete syntax of the DSL.
>
> RAG-based approaches rely on the source and target programs having similar representations in the encoding space. However, in the context of VL, the source and target languages often have completely different syntactic structures. For example, in the tensor processing domain, the source programs are written in vanilla C++, while the target language involves tensor operations like tensor-tensor arithmetic and tensor-scalar arithmetic. Moreover, multiple lines of code in the source language can be mapped to a single operator in the target (e.g., a loop adding corresponding elements of two vectors can be mapped to element-wise addition operator), and lifting-based transpilation is rarely one-to-one token-wise between the source and target languages. This significant syntactic difference makes it challenging to use a RAG-based implementation for generating a program summary in the DSL.
>
> That said, we believe RAG could be beneficial in LLMLift in a different way—specifically, in selecting examples for few-shot prompting. RAG could be used to retrieve the most relevant examples for a new program, which could then be used to prompt the model. This approach might help reduce the sample complexity of LLMLift.
>
> **For large code bases we see KV cache management to be a problem**
>
> We apologize, but we don't fully understand how KV cache management would be a significant issue in the context of VL. In the VL scenario, we're dealing with individual programs which are just a few hundred lines rather than large codebases, so the issues of KV cache management are not a concern for our current approach.
>
> **Why python as IR?**
>
> Python is the most widely represented language in the training data. This ensures that the model can generate Python code reliably with minimal prompting. Python's highly syntactic nature facilitates easier parsing. This is beneficial as a) it allows for the development of syntax-driven parsers that can efficiently translate the IR to theorem prover languages and b) It simplifies the process of translating the IR to the target DSL's concrete syntax.
>
> Direct conversion to a DSL is challenging because many DSLs may not be well-represented in the model's training data (see Figure 3). Moreover, it is also challenging to verify DSL programs using theorem provers as they do not provide support for these languages and it is not trivial to translate the DSL directly to the language supported by them. On the other hand, many theorem provers have existing libraries for handling Python-like syntax, making the verification step of LLMLift easier.
>
> **Why is accuracy the right metric?**
>
> Accuracy is the most critical metric for evaluating a transpiler, as the translated program must be semantically equivalent (checked by a formal verifier) to the given source program.  Another important metric is the performance of the translated code. Currently, the code generated by LLMLift significantly outperforms its corresponding program in the source language across all domains.
>
> **Why 0.7 as the temperature?**
>
> At lower temperatures, these models generate more deterministic outputs, which limit their ability to explore diverse solutions. Due to budget constraints, we only experimented with temperature 0.7, which is close to the optimal temperature found for code generation in recent paper[1].
>
> ### References
> [1] Evaluating Large Language Models Trained on Code. https://arxiv.org/abs/2107.03374

---

### Author Rebuttal · Authors · 2024-08-07

We thank the reviewers for their helpful comments and suggestions. We will incorporate all suggestions and clarify the confusion in our next version. Below, we address some of the common  concerns that the reviewers raised.

**How do you verify the equivalence of the source program and the program summary?**

We use Floyd-Hoare Logic (FHL) to establish the validity of generated programs[1]. In FHL, verification problem is represented as a Hoare triple $\{A\} \, P \, \{B\}$, where:
- $A$ is the pre-condition
- $P$ is the program to be executed
- $B$ is the post-condition

To establish the validity of a Hoare triple, we prove that all executions starting from states satisfying $A$, after executing program $P$, result in states satisfying $B$. This involves finding a Boolean predicate called the verification condition ($VC$) that characterizes the set of pre-conditions from which every execution of $P$ leads to a state satisfying $B$. Formally, we need to prove that the $VC$ is true given pre-condition $A$, i.e., $A \rightarrow VC(P, B)$.


Standard techniques exist to generate verification conditions from a given source program[2]. For programs containing loops, an additional predicate called a loop invariant is required. This invariant helps prove that the post-condition remains valid regardless of the number of loop iterations. The inference rules provided by FHL can be encoded into a format that can be fed into automated theorem provers or SMT (Satisfiability Modulo Theories) solvers. This encoding allows for the mechanical checking of any Hoare triple's validity.

Following are the VCs generated for our running example (Figure 1a) in the paper:
_Verification conditions:_
1. Initial: $$Inv(i=0, sum=0, data)$$
2. Preservation:

$$
\begin{gather}
Inv(i, sum, data) \land (i < size(data))
\rightarrow \\\\
Inv(i+1, ite(data[i] < 100, sum = sum +  data[i], sum = sum), data)
\end{gather}
$$

3. Termination:
$$
\begin{gather}
Inv(i, sum, data) \land \lnot(i < size(data)) \rightarrow \\\\
PS(sum, data)
\end{gather}
$$


_Generated program summary ($PS$) and $Inv$:_

\begin{align}
S &= sum \\\\
  &= reduce(map(data, \lambda j : ite(j < 100, i, 0)), \lambda a, b : a + b) \\\\
Inv &= (i \geq 0) \land (i \leq size(data)) \land sum \\\\
    &= reduce(map(data[:i], \lambda j : ite(j < 100, j, 0)), \lambda a, b : a + b) \\\\
\end{align}


_Proof:_
1. Initial Condition:
Before the loop executes, $i = 0$ and $sum = 0$. The invariant expresses sum as the result of a map followed by a reduce operation over the first $i$ elements. Since $i = 0$, the map-reduce operation is applied to an empty list, resulting in a zero sum. Therefore, the invariant holds in the initial state.
2. Preservation Condition:
The preservation condition ensures that the invariant holds throughout all iterations of the loop. This can be shown by induction. Assume the invariant holds at the $i$-th iteration. In the $(i + 1)$-th iteration, map-reduce would compute the sum for the first $i + 1$ elements, incrementing sum with the $(i + 1)$-th element of the input list data if it is less than 100.
3. Termination Condition:
The termination condition requires that the invariant implies the $PS$. When the loop terminates, $i = size(data)$, and both the $PS$ and $Inv$ expressions for sum will be identical, meaning the postcondition is satisfied.

**Why is there no performance comparison given in the experimental evaluation?**

The primary objective of verified lifting is to generate semantically equivalent programs in the target DSL from the source. DSLs are inherently designed to offer domain-specific optimizations, and the performance gains observed post-translation are attributable to the implementation of operators within the DSL rather than the translation process itself.

In LLMLift, our aim was to replicate the existing VL-based compilers. We performed a manual verification of LLMLift's output against the corresponding symbolic tools, confirming output equivalence of the two tools. Given this equivalence, the performance gains reported by the original symbolic tools are directly applicable to LLMLift's translations. Performance numbers for some of the domains are following:
1. Tensor Processing: The objective in this domain is to generate tensor programs executable on tensor processing backends. Translations to this intermediate representation (IR) yield performance gains of 2.1x (NumPy) and 167.71x (PyTorch) compared to sequential C++ implementations when compiled with GCC -O3.
2. TACO: The TACO compiler generates optimized GPU code. Translating programs to the TACO IR results in a performance gain of 24x compared to sequential C++ programs when compiled with GCC -O3.
3. Distributed Computing: The generated Spark implementations achieved an average speed-up of 15.6x compared to the sequential Java implementations. Additionally, when compared to manually written code by an expert, the generated outputs performed competitively. For more details on the user study, we refer the reader to the paper[1].

It is important to note that finding a program with optimal performance on the target backend would require performing the search phase with specific cost (objective) functions. While finding an equivalent program in the target DSL is already a challenging task, incorporating an optimization function into the search adds another layer of complexity. In addition, defining these cost functions is non-trivial in itself, as they must accurately capture the performance characteristics of the target backend. Currently, even without using cost functions, LLMLift is still able to generate performant code, as described earlier.

### References
[1] C. A. R. Hoare. An axiomatic basis for computer programming.

[2] Mike Barnett et al. Weakest-precondition of unstructured programs.

---

### Decision · Program_Chairs · 2024-09-25

**Decision:**

Accept (poster)

**Comment:**

The authors propose LLMLift: a technique using LLMs toward verified lifting. First, the LLM transforms the source program $S$ into a program summary $PS$ (in Python with the constraint that it only uses provided Python functions representing the DSL). The LLM then predicts constraints (like loop invariants $Inv$) which are passed to an automated theorem prover which attempts to prove that $PS$ is equivalent to $S$. Once verified, since $PS$ only uses DSL operators expressed as Python functions, it can be transformed with straightforward rewrite rules into a program $T$ in the target DSL. LLMLift is shown to outperform prior approaches in 4 DSLs.

Reviewers had different opinions about this paper and were not convinced by each other during discussion.
* For strengths, reviewers mentioned the problem was "timely" and "important"; LLMLift is "highly innovative", a "novel approach", and "independent and self-contained"; the approach "improves user readability" and "requires no retraining"; and the evaluation is "thorough" on a "diverse set of four scenarios" and "fully demonstrates the capability of the method".
* For weaknesses, reviewers mentioned limited novelty in combining transpilation and producing proof annotations which have been explored separately, syntactically poor writing quality, missing details about the approach and experiment results, concerns about LoC metrics to measure engineering effort, and a desire for even greater performance increase.
* The author rebuttals provide important clarifications about the missing details (verifying equivalence of $S$ and $PS$, transforming $PS$ into $T$, benchmark difficulty, reasoning behind design choices in the approach) and extra experiment results (performance of generated code, insufficiency of models weaker than GPT-4, stats on accuracy of the two steps). Although reviewers’ opinions generally did not change based on the rebuttals, I think the content of the rebuttals would greatly improve the paper if included.

Because reviewers took very different stances and did not agree, I read the entire paper myself. I wrote down my opinions and some notes as if I were providing a new review, shown after the horizontal line below.

Here are my opinions about the strengths and weaknesses listed above:
* I agree with all of the strengths.
* Although the novelty is somewhat limited, I still think it is sufficient because this is the first work to apply LLMs to transpilation and verification in one end-to-end system. The most important design consideration here is the choice of IR which enables the LLM to directly predict of $PS$, generate $Inv$, and statically transform $PS$ into $T$. I don’t think it was obvious a priori that this choice would be as effective as it was, so this counts toward novelty in my view.
* I agree that the writing quality does not yet meet the bar for publication. However, I am hopeful that the authors can improve this, especially since the rebuttals seem to have much more polished writing.
* The paper was indeed missing important details, but they have now been provided in the rebuttals.
* I do think the LoC comparisons are suspect since some details are still missing and it is hard to make the comparisons fair. Perhaps this can be resolved with more details, or reworded to be more conservative in claims. But either way, the LoC comparisons are minor compared to the main experiment results of solving more benchmarks in less time.
* In contrast to some reviewers’ opinions, I don’t think the amount of engineering for LLMLift should include any amount of the engineering of GPT-4, because GPT-4 already exists with many use-cases beyond LLMLift. It is normal and expected to build upon general-purpose tools.
* Although the reviewers provide reasonable suggestions for further experiments, I think LLMLift’s performance increase over prior approaches is already sufficient for publication.

I think the technical contribution of this paper meets the bar for publication at NeurIPS, so I recommend to **accept**. _The only concern in my view is about the writing_: there are numerous grammatical and formatting issues, and missing details must be incorporated from the rebuttals into the paper.

**To the authors:** The paper's writing very clearly needs improvement, both syntactically (grammar and formatting) and semantically (covering all important details). As it is right now, the paper's writing quality is _well below the bar for publication_. But, I recognize that you (authors) have the _ability_ to improve the writing. Specifically, it is immediately obvious that the rebuttals are written much better and contain important details that were missing from the paper. If the paper could reach a similar level of polish and detail, it would be significantly improved. I also recognize your engagement during the discussion period with lengthy rebuttals and new experiments/analyses, so I feel that you are invested in improving how your work is perceived. So, by recommending acceptance, I am generously providing you an opportunity to improve the paper without needing another round of review at a later conference. **Please do not take this for granted.** If the paper is accepted, **it is necessary to spend time adding details and improving the writing**, consulting a native English speaker and someone proficient in LaTeX as needed.

---
# My "review" of the paper

**Strengths:**
* _Most important_: there are impressive results on multiple domains compared to multiple prior approaches
  * I would not have expected an LLM to perform these tasks (generating semantically-correct Python subject to the constraint of only using provided operations, and generating invariants sufficient for formal verification) as successfully as was demonstrated, so the experimental results seem very impressive to me.
  * LLMLift outperforming multiple prior approaches on their own benchmarks is very compelling to me, especially if those benchmarks highlight the strengths of the prior approaches and not LLMLift.
  * But, contextualizing these results is very important. I suggest (below) that the authors do this by providing end-to-end examples of LLMLift’s predictions on real benchmarks.
* Verified lifting is an important topic
* Using LLMs in the proposed way is intuitive

**Weaknesses:**
* _Most important_: writing quality needs much improvement
* Important details/examples are missing
  * The rebuttals significantly reduce this weakness. The remaining portion of this weakness is the lack of end-to-end examples of LLMLift compared to prior approaches, and the uncertainty around whether details/examples will be incorporated into the paper with clear writing.
* Directly asking an LLM to perform these tasks is not particularly novel without clever prompting strategies

Questions:
* Line 79: Where does the 1000x LoC comparison come from? Section 4 mentions LoC used in prior approaches for specific goals, but we additionally need info about the total LoC (for prior approaches and LLMLift including prompt templates).
* Line 223: why zero-shot for PS and one-shot for Inv, after praising the few-shot abilities of LLMs (line 194)?
  * The reasoning provided in the rebuttal to ZcXM is satisfactory, but should be added to the paper
* Readers need more details about proving correctness and translating PS into target program
  * Extra details provided in rebuttals are satisfactory, but definitely should be added to the paper
* Table 1: The caption says “tensor processing domain” but the compared approach is C2TACO which is only mentioned in Section 4.3 (for the TACO domain). Section 4.4 (for the Tensor Processing domain) does not mention C2TACO. There must be some inconsistency.

Suggestions for improvement:
* Throughout: fix citation style and grammar issues (too numerous to list individually)
* Fig 1: this implementation of reduce seems wrong, for example, should reduce([], f) == 0? This is true for reduce with sum, but not always
* Be consistent with notation style. For example, there are at least 4 styles used for the target program T, on lines 97, 123, 124, 146.
* Line 246: == instead of =
* Appendix A: the transpile_code algorithm should be made into a numbered figure. And, `incorrect_ps_sols` and `seen_ps_sols` seem to be identical (they always contain the same elements)
* Figures 5 and 6: provide text instead of screenshots
* Table 1 is not referenced anywhere in the text, and besides, it seems not an efficient use of paper space. More important summary information can be provided instead, such as a graph or table summarizing the most important results on all 4 domains.
* It would help to provide figures (in an appendix) showing, for each of the 4 domains, a benchmark for which LLMLift outperforms the compared approaches, and the exact results (S, PS, Inv, T, generation time) produced by LLMLift and the compared approaches.

**My rating: 6** (Weak Accept: Technically solid, moderate-to-high impact paper, with no major concerns with respect to evaluation, resources, reproducibility, ethical considerations.)

In terms of technical contribution, I would say my confidence is 4/5. However, I am unsure about how much we can trust the authors to improve the writing and incorporate new details.